# Long-tailed Recognition with Model Rebalancing

**Jiaan Luo**[1,4*] **Feng Hong**[1*] **Qiang Hu**[1] **Xiaofeng Cao**[2] **Feng Liu**[3] **Jiangchao Yao**[1†]

[1]Cooperative Medianet Innovation Center, Shanghai Jiao Tong University
[2]School of Computer Science and Technology, Tongji University
[3]School of Computing and Information Systems, The University of Melbourne
[4]Shanghai Artificial Intelligence Laboratory
{luojiaan, feng.hong, qiang.hu, Sunarker}@sjtu.edu.cn
xiaofengcao@tongji.edu.cn
feng.liu1@unimelb.edu.au

## Abstract

Long-tailed recognition is ubiquitous and challenging in deep learning and even in the downstream finetuning of foundation models, since the skew class distribution generally prevents the model generalization to the tail classes. Despite the promise of previous methods from the perspectives of data augmentation, loss rebalancing and decoupled training etc., consistent improvement in the broad scenarios like multi-label long-tailed recognition is difficult. In this study, we dive into the essential model capacity impact under long-tailed context, and propose a novel framework, MOdel REbalancing (MORE), which mitigates imbalance by directly rebalancing the model's parameter space. Specifically, MORE introduces a low-rank parameter component to mediate the parameter space allocation guided by a tailored loss and sinusoidal reweighting schedule, but without increasing the overall model complexity or inference costs. Extensive experiments on diverse long-tailed benchmarks, spanning multi-class and multi-label tasks, demonstrate that MORE significantly improves generalization, particularly for tail classes, and effectively complements existing imbalance mitigation methods. These results highlight MORE's potential as a robust plug-and-play module in long-tailed settings. The code is available here.

## 1 Introduction

Deep learning has revolutionized numerous domains, from computer vision to large language models, with unprecedented performance largely fueled by large-scale well-curated datasets [Russakovsky et al., 2015]. However, many real-world uncurated data in diverse scenarios like medical diagnosis follows long-tailed distributions, where a small subset of dominant classes comprises the majority of samples, while numerous minority classes remain severely underrepresented [Krizhevsky et al., 2009]. Such ubiquitous imbalance presents a fundamental challenge for modern deep learning models that easily overfit high-frequency classes while exhibiting degraded performance w.r.t. low-frequency classes [Zhang et al., 2023]. It thus has been critical to explore robust methods against long-tailed challenges under multi-label, multi-class and even finetuning scenarios [Chen et al., 2025].

There are a range of methods developed to address the long-tailed recognition challenge from the perspectives of data augmentation, decoupled training [Kang et al., 2020b], loss rebalancing [Ma et al., 2023], and contrastive learning [Zhu et al., 2022, Du et al., 2024]. Despite promise in specific contexts, they struggle to deliver consistent improvements in broader and more complex scenarios.

---

[*]The first two authors contribute equally.

[†]The corresponding author is Jiangchao Yao (`Sunarker@sjtu.edu.cn`).

39th Conference on Neural Information Processing Systems (NeurIPS 2025).

For example, while logit adjustment [Menon et al., 2021] that builds the class frequency corrections, and probabilistic contrastive learning [Du et al., 2024] that rebalances feature representations are effective, they usually fail to take effect in the label coupling scenarios like multi-label long-tailed learning. For data augmentation ways [Shi et al., 2023], although proper re-sampling balances the class bias, the incurring cost is non-negligible, especially for the finetuning of large foundation models.

Different from previous perspectives, we explore a novel but more essential direction, which focuses on manipulating the model space of majorities and minorities and can be easily generalized to different scenarios efficiently. We start from an intuition that preserving proper model space for minority class in the manner of low-rank decomposition can help combat the imbalance challenge at the model level. Then, with a principled analysis of Rademacher complexity under space decomposition, we show that such a construction can actually support tightening the generalization bound of long-tailed learning [Menon et al., 2013, Wang et al., 2023], which guarantees the rationale of this new direction.

Based on the above analysis, we propose a novel approach called MOdel REbalancing (MORE), which partitions the parameter space to reserve dedicated capacity for minority classes while preventing dominance from majority classes (Eq. (1)). To guide this parameter space reallocation, we introduce a tailored discrepancy-based loss that measures the contribution of the low-rank component to the model's predictions (Eq. (6)), with class-wise weighting that encourages the low-rank component to focus on tail classes (Eq. (3)). This process is further optimized through a sinusoidal reweighting schedule that dynamically adjusts the influence of our reallocation loss throughout training—starting low to establish generalizable features (Eq. (5)). At inference time, low-rank components are fused with no additional computational overhead. The contributions are summarized as follows:

- We provide theoretical insights into the manner of model space manipulation under class imbalance (Theorem 1), demonstrating that by properly partitioning the model space for majority and minority classes, the generalization bounds of long-tailed learning can be further tightened, which enlightens a new direction to combat the long-tailed challenges at the model space level.

- We propose a novel method, MOdel REbalancing (MORE), for long-tailed recognition without increasing the overall model complexity or inference costs, which builds on low-rank parameter decomposition and designs a tailored discrepancy-based loss with sinusoidal scheduling that guides the proper space for minority classes and simultaneously safeguards the training of majority classes.

- We conduct extensive experiments across a diverse set of datasets, and the results show that MORE consistently improves long-tailed recognition in both single-label and multi-label settings, including those integrated with CLIP-based finetuning. The in-depth analysis discloses that MORE reduces the tendency to converge to saddle points, and proves the rationale of the module design.

## 2 Related Work

### 2.1 Single-Label Long-tailed Learning

For single-label long-tailed learning, there are substantial explorations in the recent years [Zhang et al., 2023]. At the data level, the researchers considered over-sampling and data mixing [Chawla et al., 2002, Zhong et al., 2021, Shi et al., 2023], while other transfer learning approaches [Yin et al., 2019, Wang et al., 2021a, Jin et al., 2023, Li et al., 2024a] aimed to enhance minority class feature space. However, limitations in synthetic data quality mainly restricted their effectiveness. At the model level, methods like decoupled frameworks [Kang et al., 2020a, Desai et al., 2021] separate feature representation learning from classifier optimization to reduce imbalance effects. Re-weighting techniques [Ma et al., 2023, Jiang et al., 2023, Hong et al., 2024, Luo et al., 2024] adjust the class importance during optimization, encouraging the model to pay more attention to underrepresented classes. Decision boundary adjustment [Cao et al., 2019, Menon et al., 2021, Li et al., 2022, Hong et al., 2023, Wang et al., 2025] makes effective by imposing class-specific margins, narrowing the performance gap between majority and minority classes. Recently, contrastive learning approaches [Zhu et al., 2022, Zhou et al., 2023b, Cui et al., 2024, Du et al., 2024, Zhou et al., 2024] show promise for long-tailed recognition by encouraging uniformly discriminative features in all classes. Fine-tuning foundation models [Shi et al., 2024, Li et al., 2024b] has also gained traction as a new paradigm, where lightweight approaches have demonstrated notable efficacy in long-tailed learning.

## 2.2 Multi-Label Long-tailed Learning

Due to the label coupling effect in multi-label long-tailed learning [Ridnik et al., 2021], the methods for single-label long-tailed learning usually cannot be directly applied [Tarekegn et al., 2021]. To address this challenge, various modeling methods have been explored. Recurrent neural networks [Wang et al., 2016, Yan et al., 2018] and graph convolutional networks [Chen et al., 2019] have been introduced to learn joint image-label embeddings that better capture label dependencies. Despite architectural advances, binary cross entropy (BCE) remains foundational due to its decomposition of multi-label tasks into class-wise binary objectives. To tackle label imbalance, distribution balanced loss [Wu et al., 2020] incorporates re-weighting based on label co-occurrence statistics, while asymmetric loss [Ridnik et al., 2021] introduces asymmetric focusing factors to treat positive and negative labels differently. Recent studies have explored AUC-based methods [Yang et al., 2021, Wang et al., 2022], offering deeper insights into domain adaptation for long-tailed problems in multi-class settings. Additionally, vision-language models like CLIP [Radford et al., 2021] have been adapted for multi-label recognition [Sun et al., 2022, Xia et al., 2023], leveraging label semantics to enhance generalization and mitigate imbalance through cross-modal supervision.

# 3 Method

## 3.1 Problem Setup

Consider a standard classification task under imbalanced data settings. Let the training dataset be denoted as $S = \bigcup_{i=1}^{N} \{(\boldsymbol{x}_i, y_i)\}$, where $|\mathcal{S}| = N$ is the total number of training samples, $\boldsymbol{x}_i \in \mathcal{X}$ is an input sample, and $y_i \in \mathcal{Y} \subseteq \{1, \ldots, C\}$ is its corresponding label from a total of $C$ classes. We denote the number of samples in each class as $\{N_1, N_2, \ldots, N_C\}$, and assume, without loss of generality, that $N_i < N_j$ for any $i < j$. In practice, the disparity in sample counts can be substantial, with $N_1 \ll N_C$, capturing the essence of long-tailed distributions commonly found in real-world datasets. The relative class proportions are represented by $\{\pi_1, \pi_2, \ldots, \pi_C\}$, where each $\pi_i = N_i/N$ reflects the empirical prior of class $i$. For multi-label classification, for each sample $\boldsymbol{x}_i \in \mathcal{X}$, $y_i \in \{0, 1\}^C$ represents its corresponding one-hot label vector, indicating the set of labels assigned to $\boldsymbol{x}_i$. Let $N_i'$ denote the total number of samples in which label $i$ appears, and let the total number of label occurrences across all samples be $N' = \sum_{i=1}^{C} N_i'$. The empirical prior for label $i$ is then defined as $\pi_i' = N_i'/N'$, representing the proportion of samples containing label $i$.

## 3.2 Space Decomposition

Prior research [Wang et al., 2023] has established class-wise fine-grained generalization bounds for the balanced risk, in which a critical factor is shrinking Rademacher complexity of the model. Intuitively, in long-tailed learning, minority classes usually suffer from the limited representational capacity due to their scarce examples, resulting in substantially higher Rademacher complexity for these classes. This raises an interesting hypothesis: *whether can we manipulate the model space for majority classes and minority classes to pursue a better generalization?*

We start our intuition by decomposing the parameter space with a low-rank decomposition technique, which is, though, similar to the well-known Low-Rank Adaptation (LoRA) [Hu et al., 2022] in the form, but fundamentally different in the optimization process. Formally, for a neural network with parameters $\theta$, comprising weight matrices $\{W_i\}_{i=1}^{M}$ across various layers, our core design lies in systematically decomposing each weight matrix into specialized components:

$$W_i = W_i^g + W_i^t = W_i^g + B_i^t A_i^t, \quad \forall i \in \{1, 2, \ldots, M\}, \tag{1}$$

where $W_i^g \in \mathbb{R}^{m_i \times k_i}$ captures generalizable knowledge primarily benefiting majority classes, $W_i^t \in \mathbb{R}^{m_i \times k_i}$ specializes in representing minority-specific patterns, $B_i \in \mathbb{R}^{m_i \times r}$ and $A_i \in \mathbb{R}^{r \times k_i}$ with rank $r < \min(m_i, k_i)$ guarantee the low-rank property of $W_i^t$. At the network level, this decomposition yields two complementary parameter subsets $\theta^g = \{W_1^g, W_2^g, \ldots, W_M^g\}$ and $\theta^t = \{W_1^t, W_2^t, \ldots, W_M^t\}$, and composes as a whole by $\theta = \theta^g \oplus \theta^t = \{W_1^g + W_1^t, \ldots, W_M^g + W_M^t\}$. In the following, we provide a theoretical proof that if we properly preserve the space $\theta^t$ for minority classes, we will have some potential to achieve a better generalization bound for long-tailed learning.

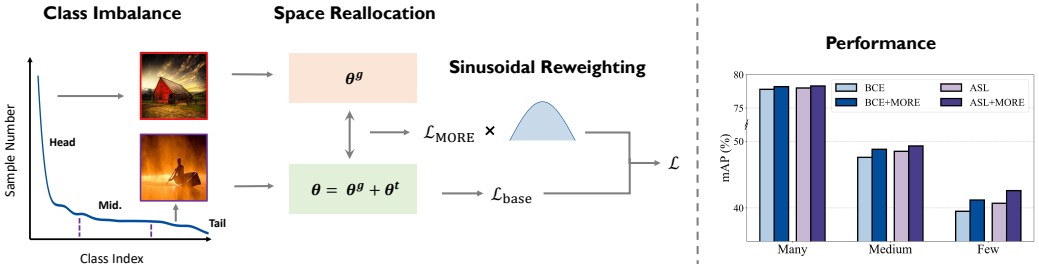

Figure 1: An overview of the proposed method's framework. The left figure illustrates how our model rebalancing is designed. The right figure presents the performance on the NUS-WIDE-SCENE dataset across the Many/Medium/Few splits. Our method demonstrates a significant improvement in the performance of minority classes, while maintaining or enhancing the performance of other classes.

### 3.3 Theoretical Understanding

**Basics.** To begin with, we provide some necessary notations and basics. For a baseline model $\mathcal{F}_0$ and our proposed model $\mathcal{F}$, where $\mathcal{F}_0 = \{f(x;\theta) \mid \theta \in \Theta\}, \Theta \subseteq \mathbb{R}^d$, and $\mathcal{F} = \{f(x;\theta^g,\theta^t) \mid \theta^g \in \Theta_g, \theta^t \in \Theta_t\}, \Theta_g \subseteq \mathbb{R}^{d_g}, \Theta_t \subseteq \mathbb{R}^{d_t}, d_t \ll d$. In long-tailed learning, standard generalization bounds fail to adequately capture performance across the class spectrum. Prior works [Ren et al., 2020, Wang et al., 2023] established class-wise generalization bounds that highlight how empirical Rademacher complexity significantly limits the generalization capabilities for minority classes. This insight motivates our further analysis in the following theorem when we manipulate the model space for majority and minority classes. For the detailed proof, please refer to Appendix B.

**Theorem 1.** *Given a function set $\mathcal{F}$, loss function $\mathcal{L}$, and training set $S$ following class-conditional distribution $D$, the balanced risk for any function $f$ is defined as $R_{bal}^{\mathcal{L}}(f) := \frac{1}{C}\sum_{y=1}^{C}\mathbb{E}_{x\sim D_y}[\mathcal{L}(f(x),y)]$. For the baseline model $\mathcal{F}_0$ and our proposed model $\mathcal{F}$ as defined above, with class proportions $\pi_y = N_y/N$ for each class $y$, the proposed model $\mathcal{F}$ enjoys a tighter generalization bound compared to the baseline $\mathcal{F}_0$. That is, for any $f \in \mathcal{F}$ and $f_0 \in \mathcal{F}_0$, it holds that $R_{bal}^{\mathcal{L}}(f) \lesssim R_{bal}^{\mathcal{L}}(f_0)$.*

**Remark 1.** By decomposing model functionality into general and minority-specific components, we separately examine their contributions to the overall complexity. Our analysis of class-specific Rademacher complexities reveals that the proposed approach redistributes modeling capacity toward minority classes while maintaining the overall complexity bound. The tighter generalization bound proves that our intuition has formal guarantees for more equitable performance across all classes.

### 3.4 Model Rebalancing

Motivated by the theoretical analysis in Section 3.3 on decomposing model parameters, we now detail the practical instantiation of parameter rebalancing through low-rank adaptation and a tailored discrepancy-based loss, which adaptively emphasizes tail-specific learning to enhance performance in long-tailed distributions.

#### 3.4.1 Parameter Space Reallocation

Given the parameter decomposition $\theta = \theta^g \oplus \theta^t$, our objective during training is to allocate $\theta^t$ for minority expertise while reserving $\theta^g$ for majority classes and general knowledge, thereby enhancing minority classes' representation through space reallocation. Remember that $f(\boldsymbol{x};\theta)$ denotes the output logits produced by parameters $\theta$ for input $\boldsymbol{x}$. Under optimal training conditions, the complete model output $f(\boldsymbol{x};\theta)$ demonstrates robust performance across the entire class distribution. Concurrently, $f(\boldsymbol{x};\theta^g)$ exhibits strong performance exclusively on majority classes while performing poorly on minority classes, confirming that $\theta^g$ successfully avoids encoding minority-specific knowledge.

To this end, we propose a tailored loss function that encourages such space reallocation by leveraging logit-level contrastive supervision. Concretely, we use a discrepancy-based method to measure the

influence of $\theta^t$ on the model's output logits. We compute the $\ell_2$ distance between the output logits of the model with $\theta^g \oplus \theta^t$ and the model with $\theta^g$:

$$\mathcal{M}(\boldsymbol{x}; \theta) = \left\| f(\boldsymbol{x}; \theta^g \oplus \theta^t) - f(\boldsymbol{x}; \theta^g) \right\|_2^2. \tag{2}$$

This discrepancy term $\mathcal{M}$ quantifies the contribution of $\theta^t$ to the model's prediction. Intuitively, this term captures the class-specific knowledge introduced by $\theta^t$ that is not present in $\theta^g$. We propose a model rebalancing loss, which encourages $\theta^t$ to learn minority expertise and $\theta^g$ to capture generalized knowledge, with a class-wise weight based on the empirical class distribution:

$$\mathcal{L}_{\mathrm{MORE}}(\theta) = \frac{1}{|\mathcal{S}|} \sum_{(\boldsymbol{x}, y) \in \mathcal{S}} \pi_y \mathcal{M}(\boldsymbol{x}; \theta). \tag{3}$$

This reweighting approach is intentionally designed to assign larger values to the majority. As a result, the loss will enforce stronger penalties for discrepancies on majority-class samples, thereby driving $\theta^t$ to minimize its influence on majority predictions. In contrast, for samples in minority classes, which have smaller weights, the loss imposes weaker penalties, allowing $\theta^t$ to retain and amplify its distinct representational contribution. This shifts the learning focus of $\theta^t$ toward minority classes, promoting effective reallocation of the model's internal capacity. In multi-label recognition, the label $y \in \{0, 1\}^C$ is a one-hot label vector. The multi-label version of $\mathcal{L}_{\mathrm{MORE}}$ is defined as follows,

$$\mathcal{L}_{\mathrm{MORE}}^{\mathrm{m}}(\theta) = \frac{1}{|\mathcal{S}|} \sum_{(\boldsymbol{x}, y) \in \mathcal{S}} \sum_{j=1}^{C} \frac{y_j}{\sum_{i=1}^{C} y_i} \pi'_j \mathcal{M}(\boldsymbol{x}; \theta^g, \theta^t), \tag{4}$$

where the summation is taken over all active labels (i.e., $y_j = 1$) to ensure that capacity reallocation remains effective and balanced across all labels, even in complex label distributions.

### 3.4.2 Sinusoidal Reweighting

To further facilitate learning through the model rebalancing loss, we introduce a dynamic weighting scheme $\alpha(\tau)$ based on a sinusoidal schedule on training time step $\tau$, which is characterized as follows,

$$\alpha(\tau) = A \cdot \sin\left(\pi \frac{\tau}{T}\right), \tag{5}$$

where $A$ is the peak amplitude controlling the maximum influence of the rebalancing loss, and $T$ is the total number of training iterations. The explanation behind this design is to gradually adjust the strength of the $\mathcal{L}_{\mathrm{MORE}}$ to balance learning priorities across different phases of optimization. At the early stage of training, the model should prioritize learning coarse-grained, easily separable representations, which are predominantly governed by majority-class samples. Therefore, we assign a small weight to the reallocation loss $\mathcal{L}_{\mathrm{MORE}}$, reducing its regularization effect and allowing $\theta^g$ to establish strong generalizable features. As training progresses into the middle phase, the weight assigned to $\mathcal{L}_{\mathrm{MORE}}$ increases, enabling the low-rank parameters $\theta^t$ to effectively learn from minority classes that are often overlooked. In the later stages, the weight is reduced again to prevent overfitting to the reallocation signal and to maintain unbiased convergence behavior.

### 3.4.3 Optimization and Inference

Our overall training framework integrates standard objective optimization with specialized space reallocation through a dynamic weighting mechanism. Given our decomposed parameter structure $\theta = \theta^g \oplus \theta^t$, the training objective is defined as follows,

$$\min_{\theta} \mathcal{L}(\theta, \tau) = \mathcal{L}_{\mathrm{base}}(\theta) + \alpha(\tau) \mathcal{L}_{\mathrm{MORE}}(\theta), \tag{6}$$

where $\mathcal{L}_{\mathrm{base}}$ denotes the primary loss function of different baseline methods, and $\mathcal{L}_{\mathrm{MORE}}$ functions as a specialized regularizer that systematically reallocates model space to protect minority classes. This dual-objective approach guides parameter optimization toward a more balanced allocation of model parameter space across majority and minority classes. Fig. 1 illustrates this training process. The pseudo-code of our training process is shown in Appendix A.

During inference, we seamlessly merge the decomposed parameters, yielding two crucial benefits: 1) Identical inference computational complexity to standard models. 2) No increase in model storage requirements. These efficiency characteristics are particularly valuable in production environments, where inference speed constitutes a primary bottleneck [Aminabadi et al., 2022].

# 4 Experiments

## 4.1 Experimental Setup

**Datasets and evaluation metrics.** We evaluate the proposed method on a suite of widely used long-tailed benchmarks, covering both single-label and multi-label image recognition settings. We conduct experiments under varying imbalance factors (IF), defined as the ratio of sample counts in the most frequent class ($N_{\max}$) to the least frequent class ($N_{\min}$). For single-label recognition, we adopt CIFAR-100-LT [Krizhevsky et al., 2009] and Places-LT [Liu et al., 2019]. CIFAR-100-LT is a long-tailed variant of the standard CIFAR-100 dataset, consisting of 100 classes with an imbalance factor of 10 and 100, where the number of samples per class follows an exponential decay. Places-LT contains 62.5k training images from 365 scene classes, with the number of samples per class varying from 5 to 4,980, and an imbalance factor of 996. For multi-label recognition, we conduct experiments on four diverse datasets: MIML [Zhou and Zhang, 2006], Pascal-VOC [Everingham et al., 2010], NUS-WIDE-SCENE [Chua et al., 2009], and MS-COCO [Lin et al., 2014]. These datasets represent a spectrum of complexity, with the number of classes ranging from 5 (MIML) to 80 (MS-COCO). The average number of labels per image varies from 1.24 (MIML) to 3.5 (MS-COCO), while the imbalance factors span from relatively balanced (1.53 for MIML) to severely imbalanced (352.92 for MS-COCO). For more information about multi-label datasets, please refer to Appendix C.1. This comprehensive evaluation suite enables us to assess our method's robustness across different multi-label recognition scenarios with varying degrees of class imbalance. We follow standard protocols in long-tailed classification by treating all classes equally during testing and reporting results across three splits: *Many*, *Medium*, and *Few*, based on the number of training samples per class. For single-label and multi-label datasets, we report top-1 accuracy and mean Average Precision (mAP) respectively as the evaluation metrics.

**Baselines.** We compare our method with a range of strong baselines commonly used in long-tailed classification. For single-label tasks, we include models trained with standard cross-entropy loss (CE), class-balanced loss (CB) [Cui et al., 2019], logit adjustment (LA) [Menon et al., 2021], balanced contrastive learning (BCL) [Zhu et al., 2022], and probabilistic contrastive learning (ProCo) [Du et al., 2024]. For multi-label classification, we evaluate against binary cross entropy (BCE), focal loss (Focal) [Lin et al., 2017], and asymmetric loss (ASL) [Ridnik et al., 2021], which are widely adopted for handling label imbalance. Additionally, since recent advances have shown the effectiveness of vision-language models in multi-label settings, we also perform experiments based on the CLIP framework, enabling a broader evaluation across modalities.

**Implementation details.** Our code is implemented with Pytorch 1.12.1. Experiments based on CIFAR-100-LT and MIML are carried out on NVIDIA GeForce RTX 3090 GPUs, while other experiments are carried out on NVIDIA A100 GPUs. For a fair comparison, we use ResNet32 on CIFAR-100-LT, ResNet34 on MIML, Pascal VOC and NUS-WIDE-SCENE, ResNet50 on ImageNet-LT and pre-trained ResNet-152 on Places-LT. We train each model with batch size of 64 (for Pascal-VOC) / 128 (for ImageNet-LT) / 256 (for CIFAR-100-LT, MIML and NUS-WIDE-SCENE) / 512 (for Places-LT) / 1024 (for MS-COCO), SGD optimizer with momentum of 0.9, weight decay of 0.0002. For multi-label tasks, the initial learning rate is set to 3e-4, with cosine learning-rate scheduling along training. For tasks based on CLIP model, we use CLIP's Transformer-based pre-trained text encoder to extract label features. During training, only vision encoder is fine-tuned, using a pre-trained ResNet34 model. Other settings are aligned with those of non-CLIP-based models.

## 4.2 Comparison Results

The efficacy of our proposed method, MORE, is assessed through comparative experiments on widely used single-label and multi-label long-tailed classification benchmarks, including finetuning pre-trained CLIP model scenarios. Detailed results are presented in Table 1 and Table 2.

**Single-label recognition.** We first evaluate MORE on CIFAR-100-LT, employing two distinct imbalance factors (IF=10 and IF=100) to test its adaptability. As evidenced in Table 1, MORE consistently elevates performance across all class frequency splits (*Many*, *Medium*, *Few*) when integrated with strong baselines like LA and ProCo. This consistent improvement, irrespective of the imbalance severity on CIFAR-100-LT, underscores the robustness of MORE and its general applicability. The performance advantages become even more critical on the large-scale Places-LT dataset, which presents a far more severe imbalance (IF = 996) and a substantially larger number of

Table 1: Top-1 accuracy (%) (↑) results for *Many*, *Medium*, *Few* and overall classes on CIFAR-100-LT and Places-LT datasets. For CIFAR-100-LT, results are categorized by imbalance factors (IF).

| Method | CIFAR-100-LT IF=10 | | | | CIFAR-100-LT IF=100 | | | | Places-LT | | | |
|---|---|---|---|---|---|---|---|---|---|---|---|---|
| | Many | Medium | Few | All | Many | Medium | Few | All | Many | Medium | Few | All |
| CE | 75.3 | 62.1 | 44.5 | 61.4 | 73.1 | 45.1 | 9.2 | 44.1 | 46.0 | 22.3 | 5.2 | 27.5 |
| CB | 66.1 | 63.5 | 55.8 | 62.1 | 72.8 | 44.8 | 11.9 | 44.7 | 46.0 | 23.6 | 10.4 | 29.1 |
| BCL | 70.7 | 62.7 | 58.5 | 64.2 | 66.8 | 52.8 | 31.9 | 51.4 | 42.4 | 41.6 | 30.4 | 39.7 |
| LA | 69.9 | 62.8 | 57.4 | 63.7 | 65.3 | 51.7 | 31.9 | 50.5 | 42.0 | 40.3 | 27.4 | 38.4 |
| +MORE | 70.9 | 64.4 | 58.2 | 64.8 | 65.3 | 52.3 | 33.6 | 51.2 | 39.5 | 42.0 | 30.6 | 39.5 |
| ProCo | 71.3 | 64.0 | 58.7 | 65.0 | 67.4 | 52.2 | 33.4 | 51.9 | 43.0 | 41.5 | 31.6 | 40.1 |
| +MORE | 72.2 | 64.4 | 60.2 | **65.9** | 68.4 | 53.5 | 34.0 | **52.9** | 43.3 | 42.2 | 33.1 | **40.8** |

classes (365). MORE continues to deliver substantial gains, particularly for the under-represented (*Medium* and *Few*) classes, which are the primary bottleneck for such severe class imbalance. Specifically, when synergistically combined with LA on Places-LT, MORE improves accuracy for *Medium* and *Few* classes by a noteworthy 1.7% and an impactful 3.2%, respectively. Collectively, these results affirm not only MORE's effectiveness in directly mitigating class imbalance and its compatibility with established rebalancing methods. For more comparison results, please refer to Appendix C.2.

**Multi-label recognition.** Our proposed method, MORE, demonstrates notable effectiveness in enhancing multi-label long-tailed classification, consistently improving performance when integrated with established baseline methods across diverse benchmarks, as detailed in Table 2. For results in Table 2 with standard deviation (Std), please refer to Table 10. The versatility of MORE is initially showcased on the MIML dataset, where its application with BCE, Focal, and ASL loss functions yields significant overall mAP gains of 3.8%, 4.0%, and 3.2%, respectively. As the dataset complexity increases, such as in PASCAL-VOC and NUS-WIDE-SCENE—both of which exhibit larger label spaces (20 and 33 classes) and more severe class imbalance (with imbalance factors of 20.92 and 159.933)—the benefits of MORE become more pronounced for under-represented classes. On PASCAL-VOC, MORE brings a substantial improvement of 3% to the *Few* mAP when combined with ASL. A similar trend is observed on NUS-WIDE-SCENE, where *Few* mAP increases by 1.7%, 1.8%, and 1.9% when MORE is applied to BCE, Focal, and ASL, respectively, without sacrificing performance on the *Many* classes. It is worth noting that in some cases, *Medium* classes may perform worse than *Few* classes. Similar observations have been made in Zhou et al. [2023b], Xia et al. [2023]. This could be attributed to the varying levels of intrinsic difficulty between classes.

**Finetuning the pretrained CLIP model.** We further evaluate the effectiveness of MORE by finetuning a pretrained CLIP on the above multi-label datasets. With the powerful pre-trained model, we observe that MORE continues to provide consistent improvements across datasets, with more pronounced gains on *Medium* and *Few* classes. As detailed in Table 2, MORE's application to the MIML dataset enhances BCE, Focal, and ASL baselines, increasing overall mAP by 1.7%, 1.8%, and 1.8%, respectively. On the more challenging PASCAL-VOC dataset, MORE combined with ASL achieves a notable 8.3% improvement for the *Few* classes, accompanied by a 3.2% gain in overall mAP. Similar trends are observed on NUS-WIDE-SCENE, where MORE enhances the performance of *Few* classes by up to 3.5%, along with a 1.5% increase in overall mAP. On the MS-COCO dataset, MORE also yields substantial gains, improving *Few* classes performance by 3.2% and overall mAP by 2.9%. These results indicate the compatibility of MORE with finetuning foundation models.

**Training overhead analysis.** We conducted additional experiments on the CIFAR-100-LT with the imbalance factor of 10 on a single NVIDIA 3090 GPU over 200 epochs. The results are shown in Table 3, which indicates that the overhead of MORE is relatively tolerable. In our implementation, we apply low-rank decomposition to all convolutional layers of the ResNet backbones, with other layers remaining unchanged. For CLIP-based models, we freeze the text encoder and fine-tune only the vision encoder and decompose all convolutional layers within it. We commonly use rank $r = 0.1$. For parameter comparisons during training: on a ResNet-34 backbone, the learnable parameters are approximately 25.4M with MORE and 21.3M without. Similarly, for ResNet-50, the counts are about 29.4M with MORE and 25.6M without. This modest training overhead stems from the low-rank parameters and is relatively contained.

Table 2: mAP (%) performance (↑) for *Many*, *Medium*, *Few*, and overall classes. Experimental evaluations conducted across four benchmarks for multi-label image recognition. Results presented for two training paradigms: training from scratch and finetuning pretrained CLIP, combining with different baseline loss functions.

| Dataset | Split | From Scratch | | | | | | Finetuning Pretrained CLIP | | | | | |
|---|---|---|---|---|---|---|---|---|---|---|---|---|---|
| | | BCE | | Focal | | ASL | | BCE | | Focal | | ASL | |
| | | / | MORE | / | MORE | / | MORE | / | MORE | / | MORE | / | MORE |
| **MIML** | Many | 85.1 | 88.8 | 84.1 | 88.4 | 84.9 | 88.4 | 96.3 | 96.4 | 95.8 | 96.6 | 96.4 | 96.8 |
| | Medium | 77.9 | 81.2 | 78.1 | 82.7 | 79.5 | 82.5 | 90.8 | 93.0 | 91.4 | 93.2 | 91.9 | 93.7 |
| | Few | 85.3 | 88.2 | 86.2 | 88.3 | 85.8 | 89.2 | 93.2 | 95.0 | 92.8 | 95.6 | 92.5 | 95.9 |
| | All | 80.8 | 84.6 | 80.9 | 85.0 | 81.8 | **85.0** | 92.4 | 94.1 | 92.6 | 94.4 | 92.9 | **94.7** |
| **PASCAL-VOC** | Many | 68.6 | 69.7 | 68.1 | 69.9 | 69.9 | 69.1 | 86.9 | 87.1 | 86.9 | 87.4 | 87.3 | 88.9 |
| | Medium | 57.6 | 59.6 | 57.7 | 60.2 | 59.0 | 60.0 | 84.0 | 84.2 | 84.5 | 84.8 | 85.1 | 87.1 |
| | Few | 52.6 | 55.0 | 53.6 | 54.1 | 52.9 | 55.8 | 81.4 | 84.5 | 83.6 | 89.1 | 82.2 | 90.5 |
| | All | 58.8 | 60.7 | 59.0 | 60.9 | 59.9 | **61.0** | 84.1 | 84.9 | 84.8 | 86.5 | 84.9 | **88.1** |
| **NUS-WIDE-SCENE** | Many | 77.8 | 78.2 | 77.6 | 78.1 | 78.0 | 78.3 | 75.1 | 73.6 | 74.3 | 74.9 | 75.6 | 74.8 |
| | Medium | 47.6 | 48.8 | 47.4 | 48.7 | 48.5 | 49.3 | 44.7 | 44.0 | 45.0 | 45.0 | 44.4 | 46.0 |
| | Few | 39.5 | 41.2 | 40.2 | 42.0 | 40.7 | 42.6 | 32.5 | 36.5 | 33.0 | 39.4 | 35.2 | 38.7 |
| | All | 54.3 | 55.4 | 54.4 | 55.6 | 55.1 | **56.1** | 50.2 | 50.7 | 50.2 | 52.3 | 51.0 | **52.5** |
| **MS-COCO** | Many | 64.2 | 65.1 | 64.5 | 65.2 | 64.9 | 65.2 | 52.9 | 50.4 | 52.1 | 50.4 | 49.0 | 51.7 |
| | Medium | 60.5 | 61.8 | 61.3 | 62.0 | 61.5 | 62.3 | 53.8 | 55.4 | 54.2 | 56.5 | 54.9 | 57.9 |
| | Few | 26.9 | 27.9 | 27.3 | 27.7 | 27.2 | 28.0 | 23.7 | 26.8 | 23.7 | 27.4 | 25.5 | 28.7 |
| | All | 57.9 | 59.1 | 58.6 | 59.3 | 58.8 | **59.6** | 51.1 | 52.5 | 51.3 | 53.4 | 51.9 | **54.8** |

Table 3: Training overhead analysis. Experiments are conducted on CIFAR-100-LT with the imbalance factor of 10 on a single NVIDIA 3090 GPU.

| Method | Training Time (Minutes) |
|---|---|
| LA | 14 |
| LA w/ MORE | 21 |
| BCL | 49 |
| ProCo | 64 |

Table 4: Analysis of $|f_y(\boldsymbol{x}; \theta^g \oplus \theta^t) - f_y(\boldsymbol{x}; \theta^g)|$ across samples from *Many*, *Medium*, *Few* classes in the final trained models on various datasets.

| Dataset | Head | Medium | Few |
|---|---|---|---|
| MIML | 0.016 | 0.018 | 0.020 |
| MIML w/ CLIP | 0.064 | 0.083 | 0.091 |
| VOC | 0.062 | 0.067 | 0.071 |
| VOC w/ CLIP | 0.026 | 0.028 | 0.036 |

**Differential impact of tail-specific parameters on logits.** To gain a deeper understanding of our mechanism, we analyze the absolute logit difference, $|f_y(\boldsymbol{x}; \theta^g \oplus \theta^t) - f_y(\boldsymbol{x}; \theta^g)|$, gauging the impact of the tail-specific parameters $\theta^t$. Table 4 shows a consistent trend across all datasets: the difference is smallest for *Head* classes and largest for *Few* classes. This indicates that $\theta^t$ responds more strongly to tail-class samples, amplifying their representations. The key insight is this relative ordering across groups, which reflects a differential effect on the logits. This pattern confirms our method effectively boosts logits for underrepresented classes.

## 4.3 Ablation Study

We perform additional ablation studies on crucial aspects of our proposed MORE: its key components and hyperparameter choices, including the reweighting schedule (Eq. (5)), peak amplitude $A$ (Fig. 2(c)), rank $r$ (Fig. 2(d)), the discrepancy metric (Eq. (2), Fig. 3(b), and Fig. 3(a)), and $\mathcal{L}_{\mathrm{MORE}}$ as a whole (Eq. (6), Fig. 3(c), and Fig. 3(d)). We validate MORE's robustness across different image resolutions (Fig. 2(a) and Fig. 2(b)). Moreover, we analyze the loss landscape (Table 6) to provide further evidence for the model rebalancing achieved.

**Ablation on different reweighting schedules (Eq. (5)).** To assess the impact of different weighting schedules on the rebalancing loss, we conduct ablation experiments using three distinct methods for the coefficient $\alpha(\tau)$: a constant setting, a cosine-based decay schedule $\alpha(\tau) = A \cdot \cos(\pi\tau/2T)$ that gradually reduces the influence of the rebalancing loss, and a sinusoidal schedule as defined in Eq. (5). These methods reflect different assumptions regarding the optimal timing and intensity of

Table 5: Top-1 accuracy (%) (↑) results on single-label dataset and mAP (%) (↑) results on multi-label datasets with different weighting schedules on $\alpha(\tau)$. Experiments conducted on single-label dataset (CIFAR-100-LT) and multi-label datasets, comparing different method combinations. *Const.* denotes constant weighting, *cos* denotes cosine-based weighting, and *sin* denotes sinusoidal weighting.

| Multi-Class Dataset | | | | Multi-Label Dataset | | | | | |
|---|---|---|---|---|---|---|---|---|---|
| Method | $\alpha(\tau)$ | IF=10 | IF=100 | Method | $\alpha(\tau)$ | MIML | VOC | NUS | COCO |
| LA+MORE | const. | 63.9 | 50.7 | BCE+MORE | const. | 82.3 | 59.3 | 54.9 | 58.2 |
| | cos | 64.0 | 50.9 | | cos | 84.0 | 59.6 | 55.2 | 58.4 |
| | sin | 64.8 | 51.2 | | sin | 84.6 | 60.7 | 55.4 | 59.1 |
| ProCo+MORE | const. | 65.2 | 52.1 | ASL+MORE | const. | 83.1 | 60.4 | 55.5 | 59.1 |
| | cos | 65.4 | 52.4 | | cos | 83.8 | 60.8 | 55.7 | 59.3 |
| | sin | **65.9** | **52.9** | | sin | **85.0** | **61.0** | **56.1** | **59.6** |

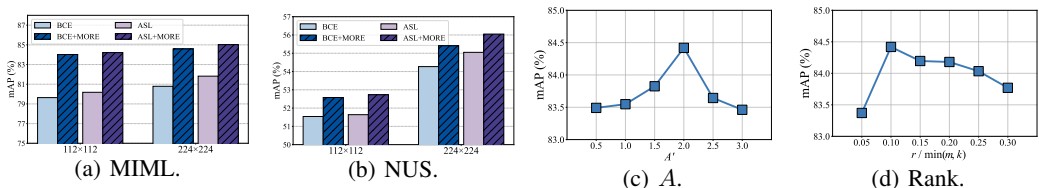

(a) MIML.  (b) NUS.  (c) $A$.  (d) Rank.

Figure 2: (a, b) mAP (%) (↑) at 112×112 and 224×224 resolutions, on MIML and NUS-WIDE-SCENE, respectively. (c) The impact of the peak amplitude $A$ in MORE. (d) The impact of the rank $r$ in MORE. In (c) and (d), experiments are conducted on MIML combined with BCE.

supervision from the rebalancing loss. We apply this ablation across both single-label and multi-label recognition settings. In Table 5, the sinusoidal method consistently improves performance over the constant and cosine-based schedules in both single-label and multi-label settings, showing benefits across all datasets. These results validate that modulating the strength of rebalancing loss over time via sinusoidal scheduling allows the model to better balance generalization.

**Ablation on peak amplitude $A$ and rank $r$.** We analyze the influence of the peak amplitude $A$ and rank the $r$ in MORE, as illustrated in Fig. 2(c) and Fig. 2(d), respectively. Due to the weight normalization in Eq. (3), $A$ exhibits sensitivity to $C$, thus we report the normalized amplitude $A' = A/C$ for clarity. Experimental results demonstrate that MORE consistently yields performance improvements across a broad spectrum of $A'$ values, with optimal performance observed at approximately 2.0. Similarly, MORE maintains robust improvement across various rank values $r$, achieving peak performance at approximately 0.1. Notably, the baseline mAP remains below 83%.

**Ablation on the discrepancy metric (Eq. (2)).** We conduct experiments to evaluate different methods for measuring distributional divergence. In our approach, we use $\ell_2$ distance to measure the discrepancy between the distributions of $f(\theta)$ and $f(\theta^g)$. For this purpose, KL divergence is also a common metric for this purpose. We compare the performance of both KL divergence and $\ell_2$ distance, as shown in Fig. 3(b) and Fig. 3(a). The results demonstrate that while KL divergence yields some improvement over the baseline, $\ell_2$ distance leads to significantly better results. This indicates that $\ell_2$ distance is a more effective measure of distributional differences in our method.

**Ablation on $\mathcal{L}_{\mathrm{MORE}}$ as a whole (in Eq. (6)).** In Fig. 3(c) and Fig. 3(d), we compare our proposed MORE with the baseline method BCE, and MORE w/o $\mathcal{L}_{\mathrm{MORE}}$ on VOC and COCO. Evidently, MORE w/o $\mathcal{L}_{\mathrm{MORE}}$ performs comparably to the baseline. However, MORE (incorporating $\mathcal{L}_{\mathrm{MORE}}$) markedly improves overall performance, with notable gains on *Few* classes. This indicates that the LoRA-like parameter decomposition (Eq. (1)) alone does not effectively alleviate class imbalance; effective mitigation is only achieved when this decomposition is combined with $\mathcal{L}_{\mathrm{MORE}}$ to realize model space rebalancing.

**Robustness across input resolutions.** We evaluate MORE under resolutions of 112×112 and 224×224 on MIML and NUS-WIDE-SCENE datasets. As shown in Fig. 2(a) and Fig. 2(b), MORE consistently improves performance across both resolutions and with both BCE and ASL losses, demonstrating robustness to different resolutions.

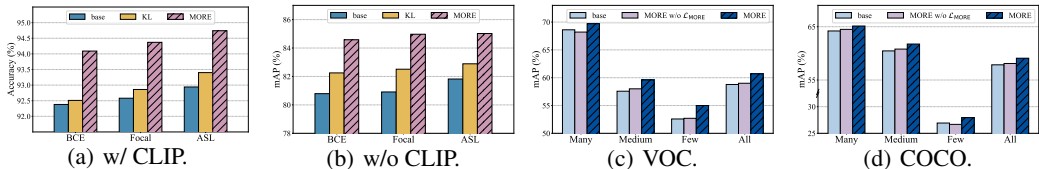

Figure 3: (a,b) mAP (%) (↑) using KL divergence and $\ell_2$ distance (MORE) as the discrepancy measure, with and without CLIP, respectively. (c,d) mAP (%) (↑) of the baseline model, MORE w/o $\mathcal{L}_{\text{MORE}}$, compared to MORE, on Pascal-VOC and MS-COCO, respectively.

Table 6: Loss landscape metrics across different methods on MIML. We report four class-wise imbalance indicators: Imb.$\lambda_{\min}$ (↓), Imb.$\lambda_{\max}$ (↓), Imb.Tr (↓), and Imb.$\gamma$ (↓), computed as the ratio between the largest and smallest absolute values of each Hessian-based metric across classes. Lower values indicate a more balanced curvature across classes. $\lambda_{min}^{(0)}$ (↑) and $\gamma_0$ (↓) represent $\lambda_{min}$ and $\gamma$ for the class with the fewest samples (class 0), where higher $\lambda_{min}$ and lower $\gamma$ suggest a flatter landscape and a reduced tendency to converge to saddle points.

| Method | CLIP | Imb.$\lambda_{min}$ | Imb.$\lambda_{max}$ | Imb.Tr | Imb.$\gamma$ | $\lambda_{min}^{(0)}$ | $\gamma_0$ |
|---|---|---|---|---|---|---|---|
| BCE | ✓ | 6.837 | 53.48 | 182.2 | 24.67 | -2181 | 0.1896 |
| BCE w/ MORE | ✓ | 5.193 | 8.303 | 11.33 | 1.612 | -597.3 | **0.0137** |
| BCE | ✗ | 13.30 | 21.90 | 23.51 | 2.832 | -3168 | 0.1257 |
| BCE w/ MORE | ✗ | **2.175** | **1.671** | **1.436** | **1.461** | **-1.932** | 0.0200 |

**Flat minima of loss landscape.** In imbalanced learning, minority classes tend to converge to saddle points in the loss landscape, which often leads to poor generalization [Dauphin et al., 2014, Zhou et al., 2023a]. Following [Rangwani et al., 2022], we focus on four metrics derived from the Hessian matrix: the minimum eigenvalue $\lambda_{\min}$, the maximum eigenvalue $\lambda_{\max}$, the eigenvalue ratio $\gamma$, and the trace of the Hessian Tr. These four metrics could indicate the sharpest directions of curvature and the overall sharpness of the landscape for each class. A low value of $\lambda_{\min}$ and a large value of $\gamma$ indicate a non-convex region and empirically suggest convergence to a saddle point, where optimization is less stable and generalization is typically weaker. To measure the degree of imbalance in the curvature across different classes, we use four class-wise imbalance indicators: Imb.$\lambda_{min}$, Imb.$\lambda_{max}$, Imb.Tr, and Imb.$\gamma$, where each is computed as the ratio between the largest and smallest absolute values of the respective metric across all classes. Table 6 demonstrates that our method substantially mitigates curvature imbalances across metrics. When combined with the MORE method, all four imbalance factors show significant reductions. Additionally, for the class with the fewest samples (class 0), the combination with MORE leads to noticeable optimizations in both $\lambda_{\min}$ and $\gamma$. These results indicate that our method not only smoothens the loss landscapes but also effectively balances the per-class curvature, particularly for underrepresented classes, thereby enhancing generalization performance. For more ablation studies, please refer to Appendix C.2.

## 5 Conclusion

In this work, we have introduced MOdel REbalancing (MORE), a novel method for addressing class imbalance by rebalancing the model space. By decomposing model parameters into main and low-rank components, MORE explicitly enhances the representation of underrepresented tail classes through a tailored loss formulation and sinusoidal reweighting approach. This approach ensures efficient and balanced learning dynamics without increasing model complexity or inference overhead. Extensive experiments across diverse long-tailed benchmarks, including both multi-class and multi-label tasks, demonstrate that MORE consistently improves performance, particularly for tail classes, and effectively integrates with existing imbalance mitigation techniques. Future work will extend MORE to other imbalanced learning scenarios, such as few-shot learning and domain adaptation, to further enhance its applicability and robustness. Further refinements can focus on mitigating representational trade-offs for semantically disparate tail classes via semantic grouping. Moreover, a stricter decoupling of general and tail-specific parameters, such as through orthogonality constraints, also offers a promising direction to enhance model modularity and reduce redundancy.

## Acknowledgement

Jiaan Luo, Feng Hong, Qiang Hu and Jiangchao Yao are supported by the National Key R&D Program of China (No. 2022ZD0160702), National Natural Science Foundation of China (No. 62306178) and STCSM (No. 22DZ2229005), 111 plan (No. BP0719010). Xiaofeng Cao is supported by National Natural Science Foundation of China (No. 62476109, No. 62206108). Feng Liu is supported by the ARC with grant number DE240101089, LP240100101, DP230101540 and the NSF&CSIRO Responsible AI program with grant number 2303037.

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

# A  Algorithm

We summarize the pseudo-code of MORE to demonstrate the procedure of implementing our method in detail, as shown in Algorithm 1.

---
**Algorithm 1** Algorithm of MORE

---
**Initialize**: $\theta^g = \{W_1^g, W_2^g, \ldots, W_M^g\}$ and $\theta^t = \{W_1^t, W_2^t, \ldots, W_M^t\}$
**for** $\tau = 1$ to $T$ **do**
    Sample mini batch $\mathcal{B} \leftarrow S$
    Calculate $\mathcal{L}_{\text{base}}$ via $f(\boldsymbol{x}; \theta^g \oplus \theta^t)$ and $y, (\boldsymbol{x}, y) \in \mathcal{B}$
    Calculate $\mathcal{L}_{\text{MORE}}$ via Eq. (3)
    Calculate $\alpha(\tau)$ via Eq. (5)
    Take gradient descent on $\nabla_{\theta^g, \theta^t}(\mathcal{L}_{\text{base}} + \alpha(\tau)\mathcal{L}_{\text{MORE}})$
    Optional: anneal the learning rate with $\tau$
**end for**

---

# B  Theoretical Supplement

In this section, we present a formal proof for Theorem 1. In imbalanced learning, the effectiveness of a model is typically evaluated based on its ability to minimize the balanced risk, which averages the risk across all classes to mitigate the impact of class imbalances. For any $f \in \mathcal{F}$, the balanced risk is defined as:

$$R_{\text{bal}}^{\mathcal{L}}(f) = \frac{1}{C}\sum_{y=1}^{C} \mathbb{E}_{x \sim D_y}[\mathcal{L}(f(x), y)], \tag{7}$$

where $C$ denotes the number of classes, $D_y$ represents the data distribution for class $y$, $\mathcal{L}$ is a loss function. Let $\mathcal{G}_0 = \{\mathcal{L} \circ f_0 : f_0 \in \mathcal{F}_0\}$ denote the hypothesis space of baseline model, and $\mathcal{G} = \{\mathcal{L} \circ f : f \in \mathcal{F}\}$ denote the hypothesis space of our proposed model.

**Lemma 1.** *Following Wang et al. [2023], for any $f \in \mathcal{F}$, the balanced risk can be bounded by the following inequality:*

$$R_{bal}^{\mathcal{L}}(f) \leq \frac{1}{C}\sum_{y=1}^{C}\hat{R}_y^{\mathcal{L}}(f) + \frac{1}{C}\sum_{y=1}^{C}\hat{\mathcal{R}}_{S_y}(\mathcal{G}) + \epsilon, \tag{8}$$

*where $\hat{R}^{\mathcal{L}}(f)$ denotes the empirical risk, $\hat{\mathcal{R}}_{S_y}(\mathcal{G})$ denotes class-specific Rademacher complexity, and $\epsilon$ is a confidence term. The class-specific Rademacher complexity is defined as:*

$$\hat{\mathcal{R}}_{S_y}(\mathcal{G}) = \mathbb{E}_\sigma\left[\sup_{g \in \mathcal{G}} \frac{1}{N_y}\sum_{i: y_i = y} \sigma_i g(x_i)\right], \tag{9}$$

*where $\sigma_i \in \{+1, -1\}$ are independent random variables.*

To prove Theorem 1, we introduce the following necessary assumptions that are formally required in the proof:

**Assumption 1.** *The hypothesis space $\mathcal{G}$ is sufficient to fit the data distribution $D$ on dataset $S$. Formally, there exist $g_{opt} = \{g_1, g_2, \ldots\}, g_i \in \mathcal{G}$ and $g_i$ that are optimal to fit $D$, so that there exists a subspace $\mathcal{G}_{sub} \subseteq \mathcal{G}$ sufficient to fit $D$. This is a common assumption, ensuring that $\mathcal{G}$ has adequate expressive power.*

**Assumption 2.** *Through parameter decomposition, the hypothesis space is partitioned as $\mathcal{G} = \mathcal{G}_{maj} \cup \mathcal{G}_{res}$, where $\mathcal{G}_{res}$ is constrained by a low-rank parameter $r$, as shown in Fig. 4(a). For an appropriate choice of $r$, $\mathcal{G}_{sub} \subseteq \mathcal{G}_{maj} \subseteq \mathcal{G}$ is sufficient to fit the majority class distribution $D_{maj}$.*

**Remark 2.** The decomposition in Assumption 2 implies that the empirical risk is approximately preserved, i.e., $\hat{R}^{\mathcal{L}}(f) \approx \hat{R}^{\mathcal{L}}(f_0)$, where $f \in \mathcal{F}$ and $f_0 \in \mathcal{F}_0$, which is generally satisfied by a proper $r$. The empirical verification presented in Fig. 4(b) supports this point inherent in Assumption 2.

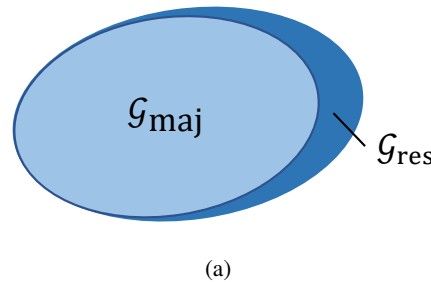
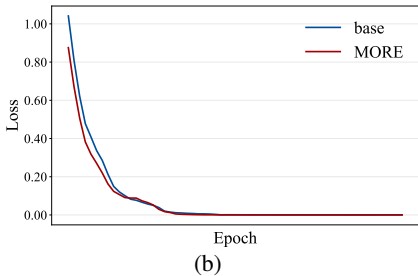

|     |     |
| :-: | :-: |
| (a) | (b) |

Figure 4: (a) Decomposition of hypothesis space $\mathcal{G}$. (b) Empirical risk comparison between baseline and MORE. Experiments are conducted on MIML using BCE as base loss.

**Proof.** The class-specific Rademacher complexity in Eq. (9) quantifies the capacity of $\mathcal{G}$ to fit random noise in the samples of each class. In our approach, we decompose $\mathcal{G}$ based on class roles; for majority classes ($y \in Y_{\text{maj}}$), the effective hypothesis space is $\mathcal{G}_{\text{maj}} \subseteq \mathcal{G}$; for minority classes ($y \in Y_{\text{min}}$), the effective hypothesis space is $\mathcal{G}_{\text{min}} = \mathcal{G}$. The risk of decomposed Rademacher complexity terms is defined as:

$$\hat{R}_1 = \sum_{y \in Y_{\text{maj}}} \hat{\mathcal{R}}_{S_y}(\mathcal{G}_{\text{maj}}), \ \ \hat{R}_2 = \sum_{y \in Y_{\text{min}}} \hat{\mathcal{R}}_{S_y}(\mathcal{G}_{\text{min}}), \ \ \sum_{y=1}^{C} \hat{\mathcal{R}}_{S_y}(\mathcal{G}) = \hat{R}_1 + \hat{R}_2. \tag{10}$$

We now demonstrate that the decomposed hypothesis space effectively reduces the overall risk of Rademacher complexity. For minority classes, $\mathcal{G}_{\text{min}} = \mathcal{G}$, which maintains identical to the baseline hypothesis space $\mathcal{G}_0$ in expressive power. Thus for any $y \in Y_{\text{min}}$, we have:

$$\hat{\mathcal{R}}_{S_y}(\mathcal{G}_{\text{min}}) = \hat{\mathcal{R}}_{S_y}(\mathcal{G}) = \hat{\mathcal{R}}_{S_y}(\mathcal{G}_0). \tag{11}$$

Summing over all minority classes, we obtain the following:

$$\hat{R}_2 = \sum_{y \in Y_{\text{min}}} \hat{\mathcal{R}}_{S_y}(\mathcal{G}_{\text{min}}) = \sum_{y \in Y_{\text{min}}} \hat{\mathcal{R}}_{S_y}(\mathcal{G}_0). \tag{12}$$

For majority classes, $\mathcal{G}_{\text{maj}} \subseteq \mathcal{G}$. For any $y \in Y_{\text{maj}}$, since the supremum is taken over a smaller set, we have:

$$\hat{\mathcal{R}}_{S_y}(\mathcal{G}_{\text{maj}}) = \mathbb{E}_\sigma \left[ \sup_{g \in \mathcal{G}_{\text{maj}}} \frac{1}{N_y} \sum_{i:y_i=y} \sigma_i g(x_i) \right] \leq \mathbb{E}_\sigma \left[ \sup_{g_0 \in \mathcal{G}_0} \frac{1}{N_y} \sum_{i:y_i=y} \sigma_i g_0(x_i) \right] = \hat{\mathcal{R}}_{S_y}(\mathcal{G}_0). \tag{13}$$

Summing over all majority classes, we obtain the following:

$$\hat{R}_1 = \sum_{y \in Y_{\text{maj}}} \hat{\mathcal{R}}_{S_y}(\mathcal{G}_{\text{maj}}) \leq \sum_{y \in Y_{\text{maj}}} \hat{\mathcal{R}}_{S_y}(\mathcal{G}_0). \tag{14}$$

By combining Eq. (10), Eq. (12) and Eq. (14), we obtain the reduced risk of Rademacher complexity:

$$\frac{1}{C} \sum_{y=1}^{C} \hat{\mathcal{R}}_{S_y}(\mathcal{G}) \leq \frac{1}{C} \sum_{y=1}^{C} \hat{\mathcal{R}}_{S_y}(\mathcal{G}_0). \tag{15}$$

Finally, by combining Eq. (15) and Eq. (8), we derive the tighter bound for the balanced risk under Assumption 1 and Assumption 2, for any $f \in \mathcal{F}$ and $f_0 \in \mathcal{F}_0$:

$$R_{\text{bal}}^{\mathcal{L}}(f) \leq R_{\text{bal}}^{\mathcal{L}}(f_0). \tag{16}$$

# C   Experimental Supplement

## C.1   Statistics of Datasets

To assess the effectiveness of the proposed approach, we perform an extensive set of experiments across four benchmark multi-label datasets, each with distinct characteristics, as outlined in Table 7. These datasets span a broad range of complexities, including variations in scale, domain, and class distribution, providing a comprehensive evaluation of the robustness and generalizability of our method. This allows us to examine the performance of our approach in multi-label recognition tasks, particularly in the presence of varying degrees of class imbalance—a challenge commonly encountered in real-world applications.

Table 7: Statistics of multi-label datasets used in our experiments. The datasets exhibit varying levels of complexity in terms of class count, training sample size, and class imbalance. The imbalance factor is calculated as the ratio between the maximum and minimum number of samples per class, with higher values indicating more severe class imbalance.

| Dataset | Classes | Samples | Min Samples/Class | Max Samples/Class | Avg. Labels/Sample | Imbalance Factor |
|---|---|---|---|---|---|---|
| MIML | 5 | 3,000 | 289 | 441 | 1.24 | 1.53 |
| Pascal-VOC 2007 | 20 | 5,000 | 96 | 2,008 | 2.5 | 20.92 |
| NUS-WIDE-SCENE | 33 | 17,500 | 75 | 11,995 | 3.4 | 159.93 |
| MS-COCO | 80 | 82,800 | 128 | 45,174 | 3.5 | 352.92 |

## C.2   More Comparison Results

**Single-label recognition.** Our approach may be conceptually analogous to Mixture-of-Experts (MoE) methods like RIDE [Wang et al., 2021b] and BalPoE [Aimar et al., 2023], as both paradigms augment model capacity for *Few* classes, albeit through fundamentally different optimization schemes. We performed a comparative analysis as shown in Table 8. The results show that integrating MORE with a strong baseline (ProCo) consistently outperforms MoE methods. Crucially, while MoE methods increase model size and computational cost with expert branches, MORE achieves these improvements with no additional inference-time parameters, highlighting its efficiency. To further verify the scalability of MORE, we evaluated our method on the large-scale ImageNet-LT dataset with a ResNet-50 backbone. As shown in Table 9, MORE provides stable improvements across all partitions, with the most significant gains on *Few* classes while maintaining performance on many and medium classes. These findings are consistent with our results on other datasets (e.g., CIFAR-100-LT, Places-LT), confirming that our method scales effectively to larger and more diverse data. Notably, the overall gains in certain settings may appear relatively modest. This may be attributed to performance saturation on well-represented classes, which restricts large improvements in the overall metric and concentrates our method's substantial gains on the more challenging *Few* classes.

Table 8: Top-1 accuracy (%) (↑) for more comparison results on CIFAR-100-LT. Results are categorized by imbalance factors (IF).

| Method | Backbone | Params | IF=10 | IF=100 |
|---|---|---|---|---|
| CE | Resnet32 | 0.57 M | 61.4 | 44.1 |
| CB | Resnet32 | 0.57 M | 62.1 | 44.7 |
| RIDE (4 experts) | Resnet32 | 1.04 M | 62.5 | 51.4 |
| BalPoE | Resnet32 | 1.37 M | 65.2 | 51.9 |
| ProCo | Resnet32 | 0.57 M | 65.0 | 51.9 |
| +MORE | Resnet32 | 0.57 M | **65.9** | **52.9** |

**Multi-label recognition.** Due to space limitations, the standard deviations for the results presented in Table 2 are not included in the main text. Results with standard deviations are provided in Table 10.

**More ablation studies.** To further analyze our method's contributions, we ablate it against two low-rank baselines on the VOC dataset: 1) BCE ($\theta^g + \theta^t$), which pairs our decomposition with

Table 9: Top-1 accuracy (%) (↑) results for *Many*, *Medium*, *Few* and overall classes on ImageNet-LT datasets.

| Method | Many | Medium | Few | All |
|--------|------|--------|-----|-----|
| CE | 69.6 | 42.2 | 14.5 | 49.0 |
| CB | 69.7 | 42.7 | 16.7 | 49.6 |
| LA | 63.7 | 51.9 | 34.7 | 54.1 |
| +MORE | 65.0 | 52.6 | 36.1 | 55.1 |
| ProCo | 66.2 | 53.9 | 37.3 | 56.3 |
| +MORE | 66.9 | 54.8 | 38.0 | **57.2** |

Table 10: mAP (%) performance metrics (↑) for overall classes. Experimental evaluations conducted across MIML, PASCAL-VOC, NUS-WIDE-SCENE, and MS-COCO benchmarks for multi-label image recognition.

| Dataset | BCE | | Focal | | ASL | |
|---------|-----|------|-------|------|-----|------|
| | / | MORE | / | MORE | / | MORE |
| MIML | $80.8_{\pm0.6}$ | $84.6_{\pm0.3}$ | $80.9_{\pm0.5}$ | $85.0_{\pm0.4}$ | $81.8_{\pm0.4}$ | $\mathbf{85.0}_{\pm0.4}$ |
| PASCAL-VOC | $58.8_{\pm0.4}$ | $60.7_{\pm0.3}$ | $59.0_{\pm0.4}$ | $60.9_{\pm0.3}$ | $59.9_{\pm0.6}$ | $\mathbf{61.0}_{\pm0.3}$ |
| NUS-WIDE-SCENE | $54.3_{\pm0.2}$ | $55.4_{\pm0.1}$ | $54.4_{\pm0.2}$ | $55.6_{\pm0.1}$ | $55.1_{\pm0.3}$ | $\mathbf{56.1}_{\pm0.2}$ |
| MS-COCO | $57.9_{\pm0.7}$ | $59.1_{\pm0.3}$ | $58.6_{\pm0.3}$ | $59.3_{\pm0.3}$ | $58.8_{\pm0.4}$ | $\mathbf{59.6}_{\pm0.2}$ |

BCE loss, and 2) BCE (LoRA), which fine-tunes with LoRA under BCE. As shown in Table 11, our method substantially outperforms both. This demonstrates that the gains stem from our overall design rather than the low-rank formulation itself. While LoRA only alters update dynamics in a fixed-capacity model, our approach rebalances model capacity to specifically counter class imbalance.

Table 11: mAP (%) performance (↑) comparison with low-rank variants.

| Method | BCE | BCE $(\theta^g + \theta^t)$ | BCE (LoRA) | BCE (MORE) |
|--------|-----|------------------------------|------------|------------|
| **Performance (%)** | 58.8 | 59.0 | 58.4 | **60.7** |

# D  Further Discussions

**Beyond a unified tail space.** Our framework represents all *Few* classes within a unified low-rank space, which may introduce representational trade-offs when certain classes are semantically disparate. While our parameter decomposition already isolates tail-specific features from the influence of *Head* classes, future work could further mitigate this intra-tail competition. One promising direction is to partition *Few* classes into semantically coherent groups (e.g., via clustering) and learn a dedicated set of low-rank parameters for each group. A simpler alternative is to increase the rank of the shared tail space to enhance its expressive capacity.

**Decoupling general and tail-specific parameters.** The interplay between the general parameters $\theta^g$, and the tail-specific parameters $\theta^t$, presents another promising direction for future investigation. Although guided by different objectives, their joint optimization may still result in representational overlap. Enforcing a stricter decoupling, for instance via an orthogonality constraint between $\theta^g$ and $\theta^t$, could yield a more modular model by minimizing this redundancy. Although certain constraints can introduce optimization challenges like impeded convergence [Vorontsov et al., 2017], exploring appropriate regularization approaches remains a valuable direction that could further improve performance on *Few* classes.

# E    Social Impact

Our research for long-tailed recognition has significant societal implications beyond its technical contributions. By addressing the fundamental challenge of class imbalance in machine learning, our work contributes to more equitable AI systems that can better serve diverse populations and use cases. Long-tailed distributions are ubiquitous in real-world scenarios, particularly in critical domains such as medical diagnosis [Dai et al., 2024, Holste et al., 2024], where rare conditions often receive inadequate representation in training data. By improving model performance on minority classes without sacrificing accuracy on majority classes, our approach helps create more reliable and fair AI systems that can recognize and respond appropriately to less common but equally important cases. The parameter space manipulation technique enables more effective learning from imbalanced datasets without requiring additional computational resources during inference. This efficiency is particularly valuable for resource-constrained environments and applications where equitable performance across all classes is essential for ethical deployment. By advancing the theoretical understanding of model space allocation in imbalanced learning scenarios and providing a practical, efficient implementation, our work contributes to the development of more inclusive AI technologies that can better serve the full spectrum of human needs, including those of underrepresented groups whose data may naturally fall into the "long tail" of many real-world distributions.

# F    Limitations, Discussions, and Future Work

Our work introduces MORE as a novel approach to long-tailed recognition through model space manipulation, demonstrating strong empirical results across diverse datasets. While effective, we acknowledge several limitations and future directions. First, our current static parameter decomposition applies uniformly across layers, whereas a dynamic decomposition strategy could adaptively allocate capacity based on layer importance for minority classes. Second, given limited training resources, we have primarily validated MORE on visual recognition tasks; extending our approach to other modalities (text, audio, video) could reveal broader applications of our parameter space manipulation principles. As foundation models continue to grow in importance, adapting MORE for extremely large-scale pre-trained models while maintaining parameter efficiency remains an exciting avenue for future exploration.

