# OpenReview forum: "Long-tailed Recognition with Model Rebalancing"
_NeurIPS.cc/2025/Conference — NeurIPS 2025 poster_

### Official Review · Reviewer_XVA7 · 2025-06-30

**Clarity:** 3
**Significance:** 3
**Originality:** 3
**Rating:** 4
**Confidence:** 4

**Summary:**

This paper introduces MORE (MOdel REbalancing), a novel method for tackling long-tailed visual recognition by rebalancing the model’s parameter space. MORE decomposes the weights of each model layer into general and minority-specific components using a low-rank parameterization, guided by a tailored discrepancy-based loss and sinusoidal reweighting schedule. This design allows the model to allocate dedicated capacity to underrepresented classes without increasing inference complexity. Theoretical analysis demonstrates that this decomposition tightens the generalization bounds. Extensive experiments across multi-class and multi-label benchmarks validate the effectiveness of the proposed method.

**Questions:**

Please see the "Strengths And Weaknesses" part.

**Ethical Concerns:**

["NO or VERY MINOR ethics concerns only"]

**Final Justification:**

I appreciate the authors’ clarifications on the large-scale benchmark evaluation, the decomposition strategy with LoRA, and the analysis of class-wise logit differences. The responses have generally addressed my concerns, and I will maintain my score of 4.

**Limitations:**

yes

**Quality:**

3

**Strengths And Weaknesses:**

**Strengths:**

- The paper is generally well-written and easy to follow.
- The proposed method is novel, addressing the long-tailed learning challenge from a new perspective by rebalancing the model's parameter space.
- The authors provide a theoretical analysis demonstrating that the proposed decomposition leads to a tighter generalization bound, which enhances the soundness and credibility of the approach.
- The method is easy to implement and can be seamlessly integrated with other long-tailed learning techniques.



**Weaknesses:**

- For the multi-class image recognition experiments, the authors primarily evaluate on CIFAR100-LT and Places-LT. However, ImageNet-LT and iNaturalist are widely used large-scale long-tailed benchmarks. Including results on these datasets would better validate the generalizability and effectiveness of the proposed method.

- The paper should be better to clarify which modules of ResNet and CLIP are decomposed using LoRA, respectively. Additionally, it would be helpful to specify the chosen rank $r$ for LoRA and compare the number of learnable parameters with and without LoRA during training.
- It would be valuable to report the values of $M(x; \theta)$ across samples from different classes in the final trained model. Furthermore, presenting the difference $f_y(x; \theta^g \oplus \theta^t) - f_y(x; \theta^g)$ across class samples would help support the effectiveness of the proposed loss $L_{\text{MORE}}$, as increasing the ground-truth logit is key to improving model performance.
- There are also some typos, such as in line 40, where "Rachemacher complexity" should be "Rademacher complexity".

---

> ### Author Rebuttal · Authors · 2025-07-31
>
> > For the multi-class image recognition experiments, the authors primarily evaluate on CIFAR100-LT and Places-LT. However, ImageNet-LT and iNaturalist are widely used large-scale long-tailed benchmarks. Including results on these datasets would better validate the generalizability and effectiveness of the proposed method.
>
> Thanks for your thoughtful recommendation. To address this, We conducted supplementary experiments on ImageNet-LT, using a ResNet-50 backbone. The accuracy (%) results are as follows:
>
> | Method        | Many  | Medium | Few   | All   |
> |---------------|:-----:|:------:|:-----:|:-----:|
> | CE            | 69.6  | 42.2   | 14.5  | 49.0  |
> | CB            | 69.7  | 42.7   | 16.7  | 49.6  |
> | LA            | 63.7  | 51.9   | 34.7  | 54.1  |
> | LA w/ MORE    | 65.0  | 52.6   | 36.1  | 55.1  |
> | ProCo         | 66.2  | 53.9   | 37.3  | 56.3  |
> | ProCo w/ MORE | 66.9  | 54.8   | 38.0  | 57.2  |
>
> These results show that MORE provides consistent and stable improvements across all partitions, with particularly notable gains on Tail classes while maintaining performance on Many and Medium classes. This aligns with our findings on CIFAR-100-LT, Places-LT and other large-scale datasets in multi-label settings, confirming the method's effectiveness in scaling to larger, more diverse datasets. Due to computational and time limits, we are currently conducting experiments on the iNaturalist dataset and will present the results as soon as they are completed.
>
> > The paper should be better to clarify which modules of ResNet and CLIP are decomposed using LoRA, respectively. Additionally, it would be helpful to specify the chosen rank $r$ for LoRA and compare the number of learnable parameters with and without LoRA during training.
>
> Thanks for this valuable suggestion. To clarify the decomposition: in our implementation, we apply low-rank decomposition to all convolutional layers of the ResNet backbones, with other layers remaining unchanged. For CLIP-based models, we freeze the text encoder and fine-tune only the vision encoder and decompose all convolutional layers within it.
>
> Regarding the rank $r$, we commonly use $r = 0.1$. For parameter comparisons during training: on a ResNet-34 backbone, the learnable parameters are approximately  25.4M with MORE and 21.3M without. Similarly, for ResNet-50, the counts are about 29.4M with MORE and 25.6M without. This modest training overhead stems from the low-rank parameters and is relatively contained. We will incorporate these details into the revised paper in Section 4.1 or appendix.
>
> > It would be valuable to report the values of $M(x;\theta)$ across samples from different classes in the final trained model. Furthermore, presenting the difference $f_y(x;\theta^g \oplus \theta^t) - f_y(x;\theta^g)$ across class samples would help support the effectiveness of the proposed loss $L_{\mathrm{MORE}$, as increasing the ground-truth logit is key to improving model performance.
>
> Thanks for this insightful suggestion. We conducted an additional analysis of $|f_y(x;\theta^g \oplus \theta^t) - f_y(x;\theta^g)|$ across samples from head, mid, and tail classes in the final trained models on various datasets. The averaged values are as follows:
>
> | Dataset      | Head   | Medium | Few    |
> | ------------ | :---:  | :---:  | :---:  |
> | MIML         | 0.016  | 0.018  | 0.020  |
> | MIML w/ CLIP | 0.064  | 0.083  | 0.091  |
> | VOC          | 0.062  | 0.067  | 0.071  |
> | VOC w/ CLIP  | 0.026  | 0.028  | 0.036  |
>
> These results suggest that $|f_y(x;\theta^g \oplus \theta^t) - f_y(x;\theta^g)|$ is smaller for head classes and larger for tail classes across all settings. This indicates that the tail-specific parameters exhibit a stronger response to tail-class samples and effectively amplify their representations. Note that the order across groups is more important than the absolute values, as it reflects their differential effect on the final logits. This pattern supports the effectiveness of our method, as it aligns with the goal of increasing ground-truth logits for underrepresented classes. We will include these analyses and visualizations in the revised paper to further substantiate the method's mechanisms.
>
> > There are also some typos, such as in line 40, where "Rachemacher complexity" should be "Rademacher complexity".
>
> Thanks for pointing this out. We have carefully proofread the entire manuscript and corrected the typos. These changes will be included in the revised version.

---

### Official Review · Reviewer_jeuY · 2025-06-30

**Clarity:** 2
**Significance:** 3
**Originality:** 3
**Rating:** 4
**Confidence:** 3

**Summary:**

This paper proposes MORE, a method designed to rebalance the model parameter space in imbalanced learning scenarios. The authors introduce a tailored loss function that incorporates logit-level contrastive supervision for space reallocation, along with a dynamic weighting scheme for model rebalancing. Experiments and ablation studies across different datasets are conducted to demonstrate the effectiveness of the proposed approach.

**Questions:**

- The connection between the theoretical analysis in Section 3.3 and the methodology in Section 3.4 is unclear. This relationship should be clarified in the rebuttal. It would also be helpful if Section 3.4 included a brief remark explicitly referencing how the theory informs or motivates the method.

**Ethical Concerns:**

["NO or VERY MINOR ethics concerns only"]

**Final Justification:**

I have read the response and other reviewers' comments. Most of my concerns have been addressed, and I maintain my positive rating for this paper.

**Limitations:**

yes

**Quality:**

3

**Strengths And Weaknesses:**

Pros

- The paper is generally well written, and the motivation is clear and well justified. The authors build on meaningful observations to guide their design.

- The proposed method is simple yet effective. Ablation studies are thorough, and the approach shows solid performance across various long-tailed learning scenarios.


Cons


- The baselines are relatively weak. The paper lacks comprehensive comparisons against recent state-of-the-art methods in long-tailed single-label classification. Including stronger baselines would help solidify the claimed improvements.

---

> ### Author Rebuttal · Authors · 2025-07-31
>
> > The baselines are relatively weak. The paper lacks comprehensive comparisons against recent state-of-the-art methods in long-tailed single-label classification. Including stronger baselines would help solidify the claimed improvements.
>
> Thank you for this valuable suggestion. Regarding mixture-of-experts (MoE) methods [1,2], we note that our approach shares conceptual similarities in addressing tail-class underfitting through capacity adjustment. To address your concern, we have included additional experiments comparing MORE against RIDE [1] (using 4 experts) and BalPoE [2] on CIFAR-100-LT, with accuracy (%) results shown below. The results demonstrate that integrating MORE with already strong baseline like ProCo yields consistent improvements. In addition to accuracy, we also report the number of inference-time parameters (third column) to highlight that, unlike MoE-based baselines which expand model capacity via expert branches, MORE requires no additional inference parameters, achieving improvements without increasing model size.
>
> | Method           | Backbone | Params | Epoch | IF=10 | IF=100 |
> | ---------------- | :------: | :----: | :---: | :---: | :----: |
> | CE               | ResNet32 | 0.57M  | 200   | 61.4  | 44.1   |
> | CB               | ResNet32 | 0.57M  | 200   | 62.1  | 44.7   |
> | RIDE (4 experts) | ResNet32 | 1.05M  | 200   | 61.8  | 49.1   |
> | BalPoE           | ResNet32 | 1.37M  | 200   | 65.1  | 52.0   |
> | ProCo            | ResNet32 | 0.57M  | 200   | 65.0  | 51.9   |
> | ProCo w/ MORE    | ResNet32 | 0.57M  | 200   | 65.9  | 52.9   |
>
> [1] Long-tailed Recognition by Routing Diverse Distribution-Aware Experts
>
> [2] Balanced Product of Calibrated Experts for Long-Tailed Recognition
>
> > The connection between the theoretical analysis in Section 3.3 and the methodology in Section 3.4 is unclear. This relationship should be clarified in the rebuttal. It would also be helpful if Section 3.4 included a brief remark explicitly referencing how the theory informs or motivates the method.
>
> Thanks for highlighting this point. To address your concern, we elaborate as follows:
>
> * **Theoretical motivation**: the theoretical foundation in Section 3.3 motivates and justifies the core components of our MORE method. In particular, Theorem 1 demonstrates that by decomposing the model parameters based on class roles, we partition the hypothesis space. This decomposition reduces the overall Rademacher complexity of the model, which in turn tightens the generalization bound on the balanced risk, particularly benefiting tail classes in long-tailed distributions.
>
> * **Methodological realization**: Section 3.4 operationalizes the above theoretical insights into a concrete implementation. Specifically, (1) The low-rank parameter decomposition reflects the model-space partition discussed in Section 3.3. (2) The discrepancy-based loss guides the optimization such that the low-rank components learn tail-specific representations, while the shared components capture general patterns. (3) The sinusoidal reweighting schedule further balances the trade-off between generalization and specialization, aligning with the theoretical goal of capacity reallocation over training time.
>
> To make this connection clearer, we will add a brief remark at the beginning of Section 3.4 explicitly referencing Section 3.3 as followes: Motivated by the theoretical analysis in Section 3.3 on decomposing model parameters, we now detail the practical instantiation of parameter rebalancing through low-rank adaptation and a tailored discrepancy-based loss, which adaptively emphasizes tail-specific learning to enhance performance in long-tailed distributions.

---

> > ### Comment · Reviewer_jeuY · 2025-08-08
> >
> > Thanks for the authors’ response. The authors have addressed my concerns. I have read the other reviewers’ comments and decided to keep my rating as borderline accept.

---

> > > ### Author Response · Authors · 2025-08-08
> > >
> > > Dear Reviewer jeuY,
> > >
> > > Thank you for taking the time to review our responses. We’re glad that our clarifications addressed your concerns. Your constructive feedback has been very helpful in improving the presentation and strengthening the overall contribution, and we will carefully incorporate these improvements in the revision. We truly appreciate your time and support.
> > >
> > > Best,
> > >
> > > The authors of Submission 10376

---

### Official Review · Reviewer_XTwi · 2025-07-01

**Clarity:** 3
**Significance:** 3
**Originality:** 3
**Rating:** 4
**Confidence:** 3

**Summary:**

This paper introduces ***MORE*** (MOdel REbalancing). The core idea is to rebalance the model's parameter space through a low-rank parameter decomposition and incorporates a difference-based loss function and a sinusoidal re-weighting schedule to achieve this. Experimental results show that ***MORE*** significantly improves the model's generalization on tail classes across several multi-class and multi-label long-tailed benchmarks.

**Questions:**

See Weaknesses.

**Ethical Concerns:**

["NO or VERY MINOR ethics concerns only"]

**Final Justification:**

The authors have satisfactorily addressed my initial concerns regarding training overhead and hyper-parameter sensitivity in their rebuttal, providing a detailed analysis of the issues I raised. Thus, I will maintain my original score.

**Limitations:**

Yes

**Quality:**

3

**Strengths And Weaknesses:**

Strengths:

1. The design of the $L_{more}$ loss is particularly insightful. By penalizing the activation of the tail-specific parameters ($θ^t$) on head-class samples, the framework effectively incentivizes these parameters to specialize in learning features pertinent to the tail classes. This is a clever and direct mechanism to encourage parameter specialization.

2. The sinusoidal re-weighting schedule is a simple yet elegant mechanism. It effectively creates a dynamic curriculum, allowing the model to alternate its focus between learning generalizable features and specialized features (for tail classes) at different training stages. This simple periodic function achieves an effect similar to more complex, hand-crafted curriculum learning strategies.

3. The use of low-rank parameterization for the tail-specific parameters is well-justified. This design choice significantly reduces the number of parameters dedicated to the tail classes, which inherently acts as a strong regularizer.

Weaknesses:

1. While the paper correctly claims no increase in inference cost, the impact on training cost is not addressed. The formulation, particularly as implied by Equation (2), suggests that the training process requires two forward passes per iteration. The lack of empirical data (GPU hours) on this training overhead is a significant omission. This information is crucial for assessing the practical feasibility of the method, especially for training on large-scale datasets.

2. I have some reservations about the sensitivity of the method to its key hyperparameters (the rank r and the balancing coefficient α). A critical trade-off exists for the rank r: if it is too small, the representational capacity of the tail-specific parameters $θ^t$ may be insufficient; conversely, if it is too large, the benefits of the low-rank assumption are diminished. It is unclear how these hyperparameters should be set for new datasets or different backbone architectures without extensive tuning. Furthermore, the assumption that a single shared low-rank space $θ^t$ can effectively serve all tail classes might be a limitation. If certain tail classes are semantically distinct and require features that are largely "orthogonal" to those of other tail classes, this shared space may fail to adequately capture their unique characteristics.

3. The paper could further explore the relationship between the general ($θ^g$) and tail-specific ($θ^t$) parameters. While the loss function applies them to different objectives, they are still optimized jointly. Have the authors considered enforcing a stricter decoupling, for instance, through an orthogonality constraint between $θ^g$ and $θ^t$? This could prevent representational overlap and lead to a more modular model.

---

> ### Author Rebuttal · Authors · 2025-07-31
>
> > While the paper correctly claims no increase in inference cost, the impact on training cost is not addressed. The formulation, particularly as implied by Equation (2), suggests that the training process requires two forward passes per iteration. The lack of empirical data (GPU hours) on this training overhead is a significant omission. This information is crucial for assessing the practical feasibility of the method, especially for training on large-scale datasets.
>
> Thanks for your comment. To address your concern, we conducted additional experiments on the CIFAR-100-LT with the imbalance ratio of 10 on a single NVIDIA 3090 GPU over 200 epochs. The results are shown in the table below, which indicates that the overhead is relatively tolerable. We will include these details in the final paper for enhanced transparency.
>
> | Method      | Training time (Minutes) |
> | ------      | :---------------------: |
> | LA          | 14                      |
> | LA w/ MORE  | 21                      |
> | BCL         | 49                      |
> | ProCo       | 64                      |
>
>
> > I have some reservations about the sensitivity of the method to its key hyperparameters (the rank r and the balancing coefficient α). A critical trade-off exists for the rank r: if it is too small, the representational capacity of the tail-specific parameters $\theta^t$ may be insufficient; conversely, if it is too large, the benefits of the low-rank assumption are diminished. It is unclear how these hyperparameters should be set for new datasets or different backbone architectures without extensive tuning. Furthermore, the assumption that a single shared low-rank space $\theta^t$ can effectively serve all tail classes might be a limitation. If certain tail classes are semantically distinct and require features that are largely "orthogonal" to those of other tail classes, this shared space may fail to adequately capture their unique characteristics.
>
> Thanks for this insightful feedback. Regarding hyperparameter sensitivity, our empirical evaluations show that our method is relatively robust, with similar settings generalizing well across diverse datasets and backbones, without the need for extensive per-task tuning. Furthermore, leveraging prior characteristics of the dataset-such as class imbalance ratios or feature distributions-to guide the selection of hyperparameters like the rank $r$ remains a valuable direction for future exploration.
>
> Your point about the potential limitation merits deeper investigation. Indeed, without our parameter decomposition, semantically distinct features in tail classes would compete for representation space with other head and tail classes. By introducing the decomposition parameters, our method already mitigates competition from head classes, allowing $\theta^t$ to focus more effectively on tail characteristics. To futher address this concern, one approach is to moderately increase the rank of tail-specific parameters for greater capacity; alternatively, we can partition tail classes into multiple groups based on semantic similarity (e.g., via clustering on pre-trained embeddings) and assign group-specific low-rank parameters, thereby alleviating potential orthogonality issues among tail features.
>
> We appreciate these suggestions and will include further discussion in the revised paper to highlight potential avenues for enhancement.
>
> > The paper could further explore the relationship between the general ($\theta^g$) and tail-specific ($\theta^t$) parameters. While the loss function applies them to different objectives, they are still optimized jointly. Have the authors considered enforcing a stricter decoupling, for instance, through an orthogonality constraint between ($\theta^g$) and ($\theta^t$)? This could prevent representational overlap and lead to a more modular model.
>
> Thank you for this constructive suggestion. While certain orthogonality constraints can pose optimization difficulties—such as the ones discussed in [1], where some forms of orthogonal regularization negatively impact the convergence rate—we still believe that appropriate orthogonalization strategies could further minimize representational overlap and potentially improve performance on tail classes. This remains a valuable direction and is worth further exploration. We will include a discussion in the conclusion section of the revised paper to reflect the merit of this suggestion.
>
> [1] On Orthogonality and Learning Recurrent Networks with Long Term Dependencies

---

> > ### Comment · Reviewer_XTwi · 2025-08-06
> >
> > Thank you for the detailed clarifications. Your responses have addressed my concerns, and I will maintain my positive rating.

---

> > > ### Author Response · Authors · 2025-08-06
> > >
> > > Dear Reviewer XTwi,
> > >
> > > Thank you for taking the time to review our responses and contribute to the development of this work. We’re glad the clarifications addressed your concerns. Your insightful feedback greatly enhanced the clarity and rigor of the paper, and we will carefully incorporate the corresponding discussions in the revision. We sincerely appreciate your support.
> > >
> > > Best,
> > >
> > > The authors of Submission 10376

---

### Official Review · Reviewer_SoG6 · 2025-07-01

**Clarity:** 3
**Significance:** 2
**Originality:** 2
**Rating:** 4
**Confidence:** 4

**Summary:**

This paper proposes MOdel REbalancing (MORE), an approach to long-tailed recognition that rebalances the model’s parameter space instead of data or loss. By decomposing model weights into general and low-rank components, MORE allocates capacity specifically for tail classes. A discrepancy-based loss and sinusoidal reweighting guide this process during training. The method improves generalization for underrepresented classes without increasing inference cost, and achieves strong results across both single- and multi-label long-tailed benchmarks, including CLIP finetuning.

**Questions:**

1. Can the authors clarify how their method is conceptually and empirically different from existing low-rank adaptation techniques? What evidence shows that this difference is critical in the long-tailed setting?

2. Can the authors evaluate the method on larger and more standard long-tailed benchmarks to better demonstrate its scalability and generalization?

3. Can the authors include comparisons or ablations against alternative capacity-based baselines to isolate the source of performance improvements?

4. How consistent are the improvements across different datasets, and can the authors explain the relatively modest gains observed in some settings?

**Ethical Concerns:**

["NO or VERY MINOR ethics concerns only"]

**Final Justification:**

Part of my concerns about the theoretical and experimental aspects has been addressed.

**Limitations:**

Yes.

**Quality:**

2

**Strengths And Weaknesses:**

**Strengths**

The method does not introduce additional inference cost, making it appealing for real-world deployment. Besides, MORE is plug-and-play and can be combined with existing reweighting or contrastive learning methods.

**Weaknesses**

1. Lack of conceptual differentiation from LoRA. Although the authors claim their low-rank decomposition is “fundamentally different” from LoRA, the distinction is not clearly articulated in terms of optimization behavior or empirical ablation. Figure 3 alone is **insufficient** to convincingly demonstrate that LoRA is ineffective in the long-tail setting. Without a sharper contrast, the contribution risks being viewed as an application of existing adaptation techniques to long-tailed learning.

 2. Evaluation breadth. Although the experiments cover a diverse range of tasks, they are limited to relatively small- or medium-scale datasets. Common benchmarks in long-tail learning—such as ImageNet-LT and iNaturalist—are notably absent. This limits the ability to draw conclusions about the method’s scalability and robustness in more realistic or challenging scenarios.

3. The paper does not explore strong baselines that manipulate model capacity, such as mixture-of-experts methods [1, 2] or self-supervised learning approaches [3], which offer alternative solutions to tail-class underfitting. Moreover, using extra parameters is a common and effective strategy—parameter-efficient finetuning approaches like [4] have also shown strong performance in such scenarios. Including comparisons or ablations against these baselines would help clarify whether the performance gains come from the specific design of MORE or from the general benefit of added parameter capacity.

4. Modest overall gains in some settings. On certain benchmarks (e.g., NUS-WIDE, CIFAR100-LT and Places-LT), the overall gains are relatively small , raising questions about trade-offs and consistency across datasets.

[1] Long-tailed Recognition by Routing Diverse Distribution-Aware Experts

[2] Balanced Product of Calibrated Experts for Long-Tailed Recognition

[3] Generalized Parametric Contrastive Learning

[4] Parameter-Efficient Long-Tailed Recognition

---

> ### Author Rebuttal · Authors · 2025-07-31
>
> > Lack of conceptual differentiation from LoRA. Although the authors claim their low-rank decomposition is “fundamentally different” from LoRA, the distinction is not clearly articulated in terms of optimization behavior or empirical ablation. Figure 3 alone is insufficient to convincingly demonstrate that LoRA is ineffective in the long-tail setting. Without a sharper contrast, the contribution risks being viewed as an application of existing adaptation techniques to long-tailed learning.
> > Can the authors clarify how their method is conceptually and empirically different from existing low-rank adaptation techniques? What evidence shows that this difference is critical in the long-tailed setting?
>
> Thank you for your comment. To address your concerns, we elaborate on the key differences below:
>
> * **Purpose and Design Motivation**: LoRA targets efficient fine-tuning by adding low-rank updates to frozen weights, aiming to reduce computational cost. Our method, in contrast, is designed specifically to address class imbalance by explicitly decomposing parameters into general and tail-specific components, without relying on a frozen backbone.
>
> * **Optimization Behavior**: LoRA applies uniform optimization across all classes using standard losses (e.g., cross-entropy), which may favor head classes. In contrast, our method incorporates a discrepancy-based loss that prioritizes tail classes, guiding tail-specific parameters to learn tail-class representations and enabling imbalance-aware learning. This positions our contribution as a novel extension of low-rank ideas to imbalance mitigation, rather than a mere application of existing techniques.
>
> To further support these conceptual differences, we conducted additional ablations on the VOC dataset with BCE, comparing variants grounded in low-rank structures: (1) BCE ($\theta = \theta^g + \theta^t$), which adopts our decomposition strategy but retains the standard BCE loss; and (2) BCE (LoRA), which applies LoRA for incremental fine-tuning of the pre-trained model with BCE. Results show MORE method surpasses these baselines with superior performance. This is because low-rank parameters like LoRA merely change update dynamics within a fixed-size network, without increase model capacity. This indicates that the performance gains of MORE stem not from the form of low-rank decomposition, but rather from its overall design, which enables the rebalancing of model capacity to effectively counter class imbalances.
>
> | Method                               | Pretrained Backbone | Frozen Backbone | Performance (%) |
> | ------------------------------------ | :-----------------: | :-------------: | :-------------: |
> | BCE                                  |          ✘          |        ✘        |      58.8       |
> | BCE (MORE)                           |          ✘          |        ✘        |      60.7       |
> | BCE ($\theta = \theta^g + \theta^t$) |          ✘          |        ✘        |      59.0       |
> | BCE (LoRA)                           |          ✔          |        ✔        |      58.4       |
>
>
> > Evaluation breadth. Although the experiments cover a diverse range of tasks, they are limited to relatively small- or medium-scale datasets. Common benchmarks in long-tail learning—such as ImageNet-LT and iNaturalist—are notably absent. This limits the ability to draw conclusions about the method’s scalability and robustness in more realistic or challenging scenarios.
> > Can the authors evaluate the method on larger and more standard long-tailed benchmarks to better demonstrate its scalability and generalization?
>
> Thanks for your thoughtful recommendation. To address this, We conducted supplementary experiments on ImageNet-LT, using a ResNet-50 backbone. The accuracy (%) results are as follows:
>
> | Method        | Many | Medium | Few  | All  |
> | ------------- | :--: | :----: | :--: | :--: |
> | CE            | 69.6 |  42.2  | 14.5 | 49.0 |
> | CB            | 69.7 |  42.7  | 16.7 | 49.6 |
> | LA            | 63.7 |  51.9  | 34.7 | 54.1 |
> | LA w/ MORE    | 65.0 |  52.6  | 36.1 | 55.1 |
> | ProCo         | 66.2 |  53.9  | 37.3 | 56.3 |
> | ProCo w/ MORE | 66.9 |  54.8  | 38.0 | 57.2 |
>
> These results show that MORE provides consistent and stable improvements across all partitions, with particularly notable gains on Tail classes while maintaining performance on Many and Medium classes. This aligns with our findings on CIFAR-100-LT, Places-LT and other large-scale datasets in multi-label settings, confirming the method's effectiveness in scaling to larger, more diverse datasets. Due to computational resource and time limits, we are currently conducting experiments on the iNaturalist dataset and will present the results as soon as they are completed.
>
> > The paper does not explore strong baselines that manipulate model capacity, such as mixture-of-experts methods [1, 2] or self-supervised learning approaches [3], which offer alternative solutions to tail-class underfitting. Moreover, using extra parameters is a common and effective strategy—parameter-efficient finetuning approaches like [4] have also shown strong performance in such scenarios. Including comparisons or ablations against these baselines would help clarify whether the performance gains come from the specific design of MORE or from the general benefit of added parameter capacity.
> > Can the authors include comparisons or ablations against alternative capacity-based baselines to isolate the source of performance improvements?
>
> Thank you for this valuable suggestion. Regarding mixture-of-experts (MoE) methods [1,2], we note that our approach shares conceptual similarities in addressing tail-class underfitting through capacity adjustment. To address your concern, we have included additional experiments comparing MORE against RIDE [1] (using 4 experts) and BalPoE [2] on CIFAR-100-LT, with accuracy (%) results shown below. The results demonstrate that integrating MORE with a strong baseline like ProCo yields consistent improvements. In addition to accuracy, we also report the number of inference-time parameters (third column) to highlight that, unlike MoE-based baselines which expand model capacity via expert branches, MORE requires no additional inference parameters, achieving improvements without increasing model size.
>
> | Method           | Backbone | Params | Epoch | IF=10 | IF=100 |
> | ---------------- | :------: | :----: | :---: | :---: | :----: |
> | CE               | ResNet32 | 0.57M  | 200   | 61.4  | 44.1   |
> | CB               | ResNet32 | 0.57M  | 200   | 62.1  | 44.7   |
> | RIDE (4 experts) | ResNet32 | 1.05M  | 200   | 61.8  | 49.1   |
> | BalPoE           | ResNet32 | 1.37M  | 200   | 65.1  | 52.0   |
> | ProCo            | ResNet32 | 0.57M  | 200   | 65.0  | 51.9   |
> | ProCo w/ MORE    | ResNet32 | 0.57M  | 200   | 65.9  | 52.9   |
>
> Our method does not inherently involve self-supervised learning. Similarly, parameter-efficient finetuning like [4] primarily deals with freezing pretrained parameters for adaptation in single label settings, which differs from our core contribution.  To further enhance comprehensiveness, we will include additional discussions on methods [3] and [4] in the revised version of the paper.
>
> [1] Long-tailed Recognition by Routing Diverse Distribution-Aware Experts
>
> [2] Balanced Product of Calibrated Experts for Long-Tailed Recognition
>
> [3] Generalized Parametric Contrastive Learning
>
> [4] Parameter-Efficient Long-Tailed Recognition
>
> > Modest overall gains in some settings. On certain benchmarks (e.g., NUS-WIDE, CIFAR100-LT and Places-LT), the overall gains are relatively small, raising questions about trade-offs and consistency across datasets.
> > How consistent are the improvements across different datasets, and can the authors explain the relatively modest gains observed in some settings?
>
> Thanks for your feedback. To address your concerns, we provide the following elaboration:
>
> * **Consistent Improvements Across Settings**: We would like to emphasize that our method actually shows reliable enhancements across diverse datasets and settings, including both single-label and multi-label long-tailed scenarios. Previous methods almost cannot achieve such generalizability and consistency.
>
> * **Focus on Underrepresented-Class Performance**: Our gains are particularly evident in tail classes, where our mechanism helps effectively mitigate underfitting due to data imbalance. For instance, on CIFAR-100-LT, MORE improves tail-class accuracy by 1.5% (IF=10) and 1.6% (IF=100). Similarly, it boosts tail-class performance by 2.5% on Places-LT and 1.9% on NUS-WIDE-SCENE, underscoring its targeted benefits for underrepresented classes.
>
> * **Performance Saturation in Certain Settings**: Regarding the relatively modest overall gains in some settings, we are inclined to attribute to the fact that, with constrained data availability in tail classes and specific model architectures, the trained models often approach a natural performance upper bound. As this ceiling is neared, the learned features become increasingly saturated, making incremental improvements more challenging to achieve. This phenomenon is common in the settings where head-class dominance already pushes baseline accuracies relatively high, leaving limited room for global gains despite substantial tail improvements. We will expand on these insights in the discussion section of the revised paper to provide further clarity.

---

> > ### Comment · Reviewer_SoG6 · 2025-08-06
> > **Response to authors**
> >
> > I really appreciate the authors' effort in addressing my concerns. My questions about the novelty and motivation of the method, as well as its differences from LoRA, have been largely addressed. However, upon examining the additional experiments, I find the results confusing. For example, ResNet-50 with vanilla CE on ImageNet-LT achieves 49.0 in overall score, which is much higher than the baseline I'm familiar with. Additionally, RIDE (4 experts) uses more parameters than expected but achieves much less gain than anticipated. Since I'm very familiar with ImageNet-LT and RIDE, these results are puzzling to me. I would greatly appreciate it if the authors could explain the codebase or experimental settings they used to help clarify these discrepancies. I believe there may be some settings that were not clearly illustrated.

---

> > > ### Author Response · Authors · 2025-08-07
> > >
> > > Thank you for your thoughtful response and valuable insights. We would like to further address your concerns:
> > >
> > > * **Regarding performance of CE on ImageNet-LT**: To clarify, our method is primarily based on the ProCo [1] codebase. It includes not only the commonly used transformations (e.g., RandomResizedCrop, RandomHorizontalFlip) as in RIDE [2] and SAM [3], but also integrates more augmentation strategies such as geometric operations (rotation, shearing, translation) and pixel-level manipulations (e.g., solarization, posterization, histogram equalization). We believe that these differences in the data **augmentation pipeline** significantly contribute to the improved performance observed in the CE baseline.
> > >
> > > * **Regarding the performance of RIDE**: We apologize for the confusion. As RIDE, BalPoE, and ProCo all claim to follow the LDAM [4] setting on CIFAR-100-LT, our initial results were based on the official codebases of RIDE and BalPoE without realizing the detailed discrepancy. Upon re-examination, we identified their data augmentation differences from ProCo (Here, we also appreciate the reviewer's reminding). We have now fully aligned all settings under the ProCo codebase and re-run the experiments. **After alignment, RIDE shows improvements from 61.8% to 62.5% under imbalance factor of 10, and shows improvements from 49.1% to 51.4% under imbalance factor of 100, respectively**. As shown below, our method still achieves consistent improvements. Note that, if we follow the augmentation method used in the original RIDE, we find CE only achieves 39.3% under imbalance factors of 100, which further confirms the advantage of ProCo codebase. About the smaller performance gap of RIDE under our setting, we conjecture that the stronger data augmentation we adopt, may suppress the RIDE’s relative advantages. We appreciate your feedback and will update the results and include this discussion in the revision.
> > >
> > >    | Method           | Backbone | Epoch | Params | IF=10 | IF=100 |
> > >    | ---------------- | :------: | :---: | :----: | :---: | :----: |
> > >    | CE               | ResNet32 | 200   | 0.57 M | 61.4  | 44.1   |
> > >    | CB               | ResNet32 | 200   | 0.57 M | 62.1  | 44.7   |
> > >    | RIDE (4 experts) | ResNet32 | 200   | 1.04 M | 62.5  | 51.4   |
> > >    | BalPoE           | ResNet32 | 200   | 1.37 M | 65.2  | 51.9   |
> > >    | ProCo            | ResNet32 | 200   | 0.57 M | 65.0  | 51.9   |
> > >    | ProCo w/ MORE    | ResNet32 | 200   | 0.57 M | 65.9  | 52.9   |
> > >
> > > * **On parameters and FLOPs**: To avoid potential misunderstandings, we provide a comparison of RIDE with different numbers of experts in terms of FLOPs, parameter count, and accuracy (%) on CIFAR-100-LT with an imbalance ratio of 100. The table below summarizes the details for your reference. **Note that, RIDE reduces the dimensionality of layers in backbone, which results in a smaller backbone compared to those used in other methods. So RIDE (2 experts) that has 0.53M parameters, is smaller than vanilla CE baseline that has 0.57M parameters, which is not a surprise.**
> > >
> > >    | Model            | FLOPs   | Params | Performance (%) |
> > >    | ---------------- | :-----: | :----: | :-------------: |
> > >    | RIDE (2 experts) | 76.4 M  | 0.53 M | 49.1            |
> > >    | RIDE (3 experts) | 102.2 M | 0.79 M | 50.8            |
> > >    | RIDE (4 experts) | 128.0 M | 1.04 M | 51.4            |
> > >
> > > We hope this helps address your concerns. Thank you again for your constructive feedback.
> > >
> > >
> > > [1] Probabilistic Contrastive Learning for Long-Tailed Visual Recognition
> > >
> > > [2] Long-tailed Recognition by Routing Diverse Distribution-Aware Experts
> > >
> > > [3] Escaping Saddle Points for Effective Generalization Class-Imbalanced Data
> > >
> > > [4] Learning Imbalanced Datasets with Label-distribution-aware Margin Loss

---

> > > > ### Comment · Reviewer_SoG6 · 2025-08-09
> > > > **Response to authors**
> > > >
> > > > Thanks for your reply. The results you provided are detailed and convincing, and I have decided to raise the final score.

---

> > > ### Author Response · Authors · 2025-08-08
> > >
> > > Dear Reviewer SoG6,
> > >
> > > We sincerely thank you for your constructive comments. As the discussion period is drawing to a close, we kindly wanted to check whether your concerns have been addressed. We would greatly appreciate your valuable feedback.
> > >
> > > Best regards,
> > >
> > > The authors of Submission 10376

---

> ### Author Response · Authors · 2025-08-04
>
> Dear Reviewer SoG6,
>
> We sincerely thank you for your thoughtful and constructive feedback, which has significantly helped us improve our manuscript. Below we summarize our key responses:
>
> - **Differentiation from LoRA:** We distinguished our method from LoRA by clarifying their fundamental differences: our approach tackles class imbalance via parameter decomposition and discrepancy-based loss, contrasting with LoRA's uniform fine-tuning for efficiency. Additional empirical ablations further demonstrated these distinctions and our method's effectiveness in long-tailed scenarios.
>
> - **Evaluation Breadth:** We conducted additional experiments on ImageNet-LT to demonstrate our method's scalability and generalization on large-scale long-tail benchmarks. Our method consistently improved performance across all class partitions when integrated with strong baselines like LA and ProCo, confirming its broad applicability.
>
> - **Comparisons Against Capacity-Based Baselines:** We compared MORE with Mixture-of-Experts (MoE) methods like RIDE and BalPoE. Our method delivered consistent improvements without increasing inference-time parameters. This demonstrates that MORE's performance improvements stem from its specific design, rather than expanding model capacity as MoE approaches do.
>
> - **On Modest Overall Gains:** We clarified that while tail-class improvements are substantial, overall gains may appear modest because, with constrained data and specific architectures, models approach a natural performance upper bound. This limits substantial global improvements especially when head-class performance is already high.
>
> We hope that our detailed responses and the additional experiments address your concerns. Please let us know if anything is unclear. We truly value your time and would be most grateful for any feedback you could give to us.
>
> Best Regards,
>
> The authors of Submission 10376

---

> ### Author Response · Authors · 2025-08-09
> **Thanks for increasing your score!**
>
> Dear Reviewer SoG6,
>
> Thank you for taking the time to review our responses and provide your valuable feedback. We are glad that our clarifications have addressed your concerns. Your thoughtful comments have been essential in improving the clarity and rigor of our work. We will ensure that these improvements are incorporated into the revision. We sincerely appreciate your support and the constructive suggestions you have provided.
>
> Best regards,
>
> The authors of Submission 10376

---

### Comment · Area_Chair_ZVZ1 · 2025-08-05

Dear Reviewers,

Please read the other reviews and the author's response, and start a discussion with the authors promptly to allow time for an exchange.

Your AC

---

### Note · Authors · 2025-08-13

Dear reviewers and ACs,

We sincerely thank all reviewers for their constructive feedback and active engagement during the rebuttal and discussion phases. Specially thanks the Area Chair for facilitating this productive process. We appreciate that reviewers considered the key strengths of our work:

- Novel approach to long-tailed recognition by rebalancing the model’s parameter space (Reviewer SoG6, XTwi, jeuY & XVA7)
- No additional inference cost and easy integration with existing methods (Reviewer SoG6, XTwi & XVA7)
- Insightful discrepancy-based loss and elegant sinusoidal reweighting schedule (Reviewer XTwi)
- Solid theoretical analysis to improved generalization bounds (Reviewer XVA7)
- Well-written presentation and easy to follow (Reviewer jeuY & XVA7)

During the rebuttal and discussion, we have addressed all major concerns through further clarifications and experiments, with no remaining issues raised by reviewers. Specifically, we:

- Provided critical clarifications:
  - Distinguished MORE from existing low-rank adaptation approaches (SoG6 Q1)
  - Clarified hyperparameter robustness and potential extensions (XTwi W2, W3)
  - Connected theoretical results directly to the methodological design (jeuY Q1)
  - Specified decomposition details (XVA7 W2)

- Conducted extra experiments:
  - New large-scale benchmark results (SoG6 Q2; XVA7 W1)
  - Comparisons with more baselines (SoG6 Q3; jeuY W1)
  - Training cost analysis (XTwi W1)
  - Additional ablations and logit-difference analysis (XVA7 W3)

We once again express our gratitude to all reviewers for their time and effort devoted to evaluating our work. We are committed to incorporating all suggested improvements into the final version to further strengthen the paper.

Best,

The authors of Submission 10376

---

### Decision · Program_Chairs · 2025-09-17

**Decision:**

Accept (poster)

**Comment:**

This paper proposes a new method for long-tail learning by rebalancing the model’s parameter space, rather than relying on data or loss functions. By decomposing model weights into general and low-rank components, the proposed method allocates capacity specifically for tail classes. The method improves generalization for underrepresented classes without increasing inference cost, and achieves strong results across both single- and multi-label long-tailed benchmarks, including CLIP finetuning.

After rebuttal, the paper has received four positive evaluations. All reviewers have updated their final ratings and noted that the majority of their concerns in both theoretical and experimental aspects have been resolved through the author-reviewer discussion. Additionally, every reviewer has recognized the paper’s contributions and expressed satisfaction with its technical strengths.

The AC concurs with the reviewers' assessments and recommends acceptance.